# Proteomic characterization of epithelial ovarian cancer delineates molecular signatures and therapeutic targets in distinct histological subtypes

Ting-Ting Gong[1,8], Shuang Guo[2,8], Fang-Hua Liu[3,4,5], Yun-Long Huo[6], Meng Zhang[3,4,5], Shi Yan[3,4,5], Han-Xiao Zhou[2], Xu Pan[2], Xin-Yue Wang[2], He-Li Xu[3,4,5], Ye Kang[6], Yi-Zi Li[3,4,5], Xue Qin[1], Qian Xiao[1,3], Dong-Hui Huang[3,4,5], Xiao-Ying Li[3,4,5], Yue-Yang Zhao[3], Xin-Xin Zhao[3], Ya-Li Wang[3], Xiao-Xin Ma[1], Song Gao[1], Yu-Hong Zhao[3,4,5], Shang-Wei Ning [2,9] ✉ & Qi-Jun Wu [1,3,4,5,7,9] ✉

Clear cell carcinoma (CCC), endometrioid carcinoma (EC), and serous carcinoma (SC) are the major histological subtypes of epithelial ovarian cancer (EOC), whose differences in carcinogenesis are still unclear. Here, we undertake comprehensive proteomic profiling of 80 CCC, 79 EC, 80 SC, and 30 control samples. Our analysis reveals the prognostic or diagnostic value of dysregulated proteins and phosphorylation sites in important pathways. Moreover, protein co-expression network not only provides comprehensive view of biological features of each histological subtype, but also indicates potential prognostic biomarkers and progression landmarks. Notably, EOC have strong inter-tumor heterogeneity, with significantly different clinical characteristics, proteomic patterns and signaling pathway disorders in CCC, EC, and SC. Finally, we infer MPP7 protein as potential therapeutic target for SC, whose biological functions are confirmed in SC cells. Our proteomic cohort provides valuable resources for understanding molecular mechanisms and developing treatment strategies of distinct histological subtypes.

Epithelial ovarian cancer (EOC) is the most common type of ovarian cancer (OC), and is considered to be a malignant transformation from the ovarian surface, peritoneal, or fallopian tube epithelium[1]. According to the 2020 World Health Organization (WHO) classification[2], common histological subtypes of EOC include clear cell carcinoma (CCC), endometrioid carcinoma (EC), serous carcinoma (SC), and mucinous carcinoma. Among them, SC, the most prevalent EOC subtype, shows the worst survival outcome and is the leading cause for gynecological cancer-related deaths[3]. As a result, SC has received widespread attention, particularly in studies of large cohorts, such as The Cancer Genome Atlas (TCGA)[4]. Exploratory genome and transcriptome data from SC cohorts have increased our overall understanding of EOC[5,6].

[1]Department of Obstetrics and Gynecology, Shengjing Hospital of China Medical University, Shenyang, China. [2]College of Bioinformatics Science and Technology, Harbin Medical University, Harbin, China. [3]Department of Clinical Epidemiology, Shengjing Hospital of China Medical University, Shenyang, China. [4]Clinical Research Center, Shengjing Hospital of China Medical University, Shenyang, China. [5]Key Laboratory of Precision Medical Research on Major Chronic Disease, Shengjing Hospital of China Medical University, Shenyang, China. [6]Department of Pathology, Shengjing Hospital of China Medical University, Shenyang, China. [7]NHC Key Laboratory of Advanced Reproductive Medicine and Fertility (China Medical University), National Health Commission, Shenyang, China. [8]These authors contributed equally: Ting-Ting Gong, Shuang Guo. [9]These authors jointly supervised this work: Shang-Wei Ning, Qi-Jun Wu. ✉e-mail: ningsw@ems.hrbmu.edu.cn; wuqj@sj-hospital.org

As the improvement in analytical depth via mass spectrometry (MS) has proceeded, researchers have captured druggable molecular phenotypes at the proteome levels in multiple cancer types. In glioblastoma, proteomic analysis has characterized the signature of the epithelial-mesenchymal transition (EMT) as specific to tumor cells but not to stroma in the mesenchymal subtype[7]. For hepatocellular carcinoma, Jiang et al. demonstrated that sterol O-acyltransferase 1 (SOAT1), with a subtype-specific signature, suppressed the proliferation and migration of cancer cells[8]. Furthermore, several studies have explored the important role of proteomic profiling in high-grade serous ovarian cancer (HGSOC). The study by Zhang et al. provided additional insights into the pathways and processes that drive the biology of HGSOC and how these pathways are altered relative to the clinical phenotypes[9]. McDermott et al. described a potential role for proliferation-induced replication stress in promoting the characteristic chromosomal instability of HGSOC[10]. A study from Coscia et al. also revealed that cancer/testis antigen 45 (CT45), a prognostic factor associated with the doubling of disease-free survival, enhanced chemosensitivity in metastatic HGSOC[11]. However, molecular characterization of EOC histological subtypes (CCC, EC, and SC) using large cohorts is still limited.

In this work, we collect a cohort of 269 EOC samples, aiming at a comprehensive characterization of EOC based on proteomic analysis, to increase our knowledge of the molecular features associated with this lethal malignancy. A comparison of the proteomic profile of EOC and control tissue (CT) samples reveals dysregulated proteins and aberrant signaling pathways. The protein co-expression network not only reflects the biological features of each co-expression module, but also suggests potential prognostic biomarkers and progression landmarks. There is strong heterogeneity among the histological subtypes of EOC. Based on protein expression levels, prognostic ability and druggability, we also predict potential therapeutic targets for each subtype separately.

## Results

### Proteomic landscape highlights heterogeneity and differences in clinical characteristics of EOC

We collected 239 EOC samples (SC, $n = 80$; EC, $n = 79$; and CCC, $n = 80$) and 30 CT samples (Supplementary Data 1). The proteome analysis was performed using label-free technology on the same mass spectrometer with consistent quality control (Fig. 1a). A total of 8257 proteins were identified across all tumor samples. The number of proteins detected in each sample was shown in Supplementary Fig. 1a, in which Supplementary Fig. 1b demonstrated the details of the proteomic data quantification for each group. Uniform Manifold Approximation and Projection (UMAP) analysis demonstrated a clear distinction between the proteomes of tumor and non-tumor samples as well as between EOC pathology subtypes, which further highlights the high heterogeneity among tumor samples that underpins our stratification analysis (Fig. 1b). To better depict the inter-tumor heterogeneity of EOC, we characterized the patients' clinical information among pathology subtypes. Compared with EC and CCC, patients with SC were older at diagnosis and were prone to relapse (Fig. 1c and Supplementary Data 2). Moreover, SC was enriched with advanced tumor stages than EC and CCC (Fig. 1d and Supplementary Data 2). In particular, evaluation of the survival characteristics of the EOC histological subtypes revealed that SC, EC, and CCC exhibited significantly different overall survival (OS) and relapse-free survival (RFS) (Fig. 1e and Supplementary Data 2). Of these, SC had a significantly lower survival rate and a greater risk of postoperative death and recurrence, with a median OS of $47.57 \pm 2.36$ months and a median RFS of $16.57 \pm 2.50$ months (median ± standard error). Furthermore, we delineated protein subcellular localization and found that most detected proteins were from the nucleus ($n = 2834$), plasma membrane ($n = 740$), and

mitochondria ($n = 620$), implying potential protein targets and dysregulated biological processes (BPs) (Fig. 1f).

### Dysregulated proteins impact important biological processes

Dysregulated proteins in tumors were identified based on proteomic data. Of the 4447 high-quality proteins obtained after quality control, 295 and 927 proteins were significantly up- and downregulated in the tumor samples, respectively (adj.$P$ value < 0.01 and |log2 (fold change)| >1, Fig. 2a and Supplementary Data 3). Intriguingly, we found that most dysregulated proteins tended to be downregulated in tumor tissue, which was also supported by the previous study[10]. In addition, we found that upregulated proteins had higher average expression than downregulated proteins (Supplementary Fig. 2a). In addition, we calculated the mutation frequency of these dysregulated proteins in SC samples using the somatic mutation data of the TCGA mc3 project[12], and the results showed no significant difference in the mutation frequency of dysregulated proteins (Supplementary Fig. 2b).

Functional enrichment analysis was applied to dysregulated proteins with a fold change larger than 2, and GO analysis indicated that downregulated proteins were significantly enriched in BPs including: cell death, DNA repair/damage, development and immune response, and others, whereas BPs that were enriched in upregulated proteins were limited (Fig. 2b, Supplementary Fig. 2c, and Supplementary Data 4). As for KEGG pathways, DNA replication, cell cycle, HIF-1 signaling pathway and several metabolism-related pathways were overrepresented in proteins upregulated in tumor samples, whereas PI3K-Akt signaling pathway and focal adhesion and extracellular matrix (ECM)-receptor interaction were overrepresented in downregulated proteins (Fig. 2c and Supplementary Data 4). Meanwhile, the GSEA-enriched GO biological processes and KEGG pathways also highlighted consistent biological functions (Supplementary Data 5). The GSEA analysis of cancer hallmarks revealed that the estrogen response and oxidative phosphorylation hallmarks were overrepresented in the upregulated proteins, and that apoptosis, EMT, TNFA signaling via NF-κB, and other pathways were overrepresented in the downregulated proteins in EOC (Fig. 2d), which indicated that the proliferation and metastasis of tumor cells may be promoted through aberrant proteins and pathways.

We aimed to measure the ability of differentially expressed proteins to classify EOC and CT samples. Firstly, to distinguish between tumor and normal samples, differentially expressed proteins were used as the initial feature. Then, the classification performance of each feature was evaluated using the area under the curve (AUC) of the receiver operating characteristic (ROC) curves (R package "pROC"[13]). In addition, proteins with good discriminatory power were enrolled for further validation in the ovarian cancer cohort of McDermott et al. (Clinical Proteomic Tumor Analysis Consortium, CPTAC cohort)[10]. We observed that multiple upregulated proteins exhibited strong discriminatory ability with a mean AUC greater than 0.9 (current cohort) and further validated in the CPTAC cohort[10] with a mean AUC also greater than 0.8. The exosome protein lists from the ExoCarta (http://www.exocarta.org/)[14] and Vesiclepedia (http://microvesicles.org/)[15] databases. Proteins with discriminatory ability have overlapping portions with exosomal proteins (Fig. 2e), and these overlapping proteins may provide richer perspectives for EOC research.

### Signaling pathway disturbances suggest latent therapeutic opportunities

Since the proteomic analysis revealed disturbances in several signaling pathways, we next sought to elucidate such disturbances from a multi-omics perspective. Here, we integrated previously published phosphoproteomic data (CPTAC cohort)[10] and phosphoproteomic analysis-identified dysregulated phosphosites of these proteins in the above pathways. As a result, we found that proteins in DNA replication pathways showed global upregulated patterns, whereas dysregulated

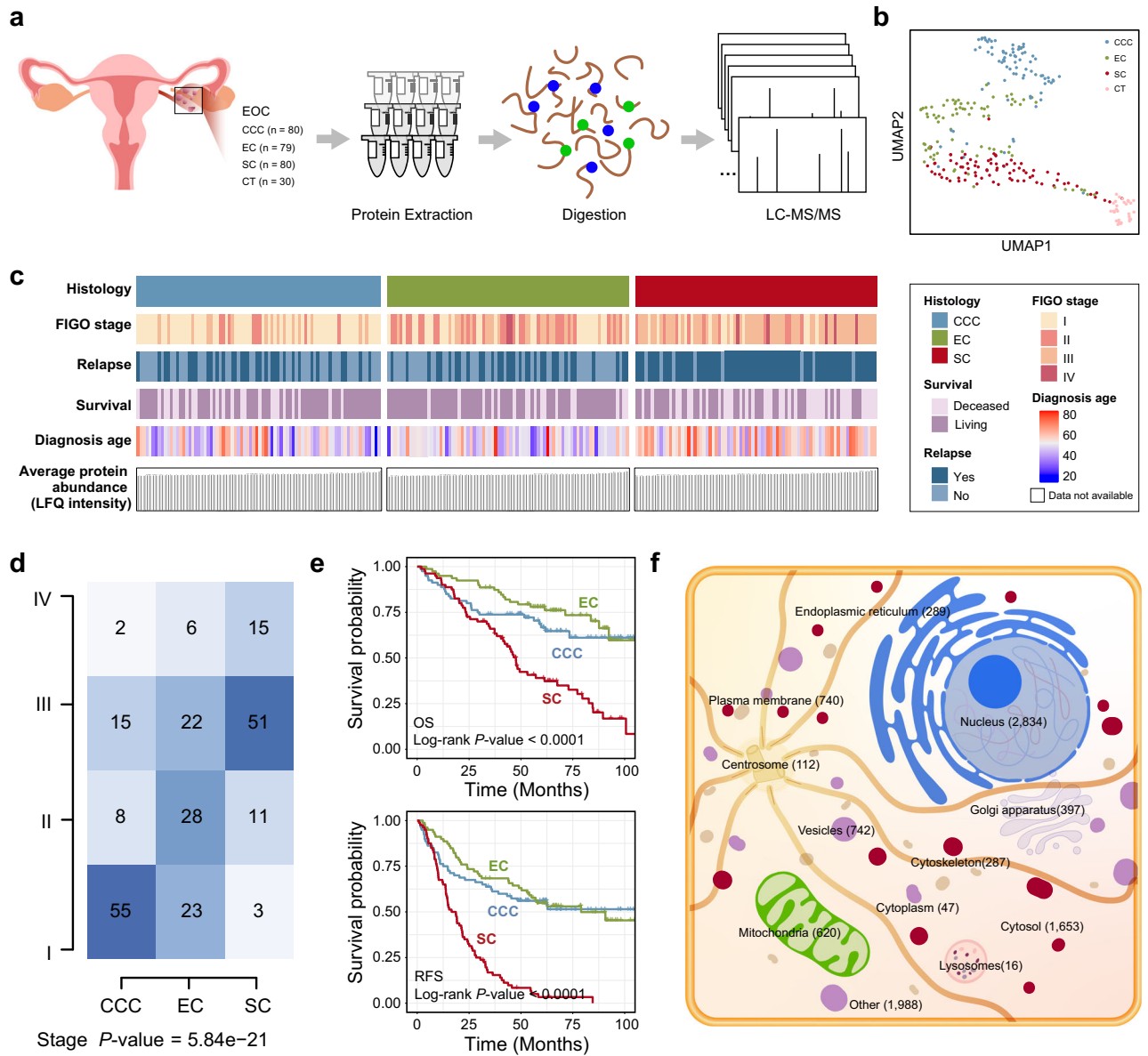

**Fig. 1 | Proteome landscape in EOC histological subtypes. a** Overview of the experimental setup for MS-based proteome profiling. **b** UMAP plot of epithelial ovarian cancer (EOC) tumor and control tissue (CT) samples, color-coded by EOC histological subtypes. **c** Heatmap showing the clinical information and mean protein abundance of samples. **d** Differences in the abundance of EOC histological subtypes in terms of tumor stage. CCC samples ($n = 80$), EC samples ($n = 79$), and SC samples ($n = 80$). $P$ values were calculated by two-sided Fisher's exact test. **e** Kaplan–Meier plots of overall survival (OS) (Log-rank $P$ value = 4e-08) and relapse-free survival (RFS) (Log-rank $P$ value < 2e-16) for EOC histological subtypes. CCC samples ($n = 80$), EC samples ($n = 79$), and SC samples ($n = 80$). **f** Schematic diagram of protein subcellular localization. Icons were made by Freepik (https://www.freepik.com/home). Source data are provided as a Source Data file.

phosphosites showed more complex patterns in Gap 1 Phase, Gap 2 Phase and Mitotic Phase. The HIF-1 signaling pathway was also enriched in upregulated proteins, whereas the phosphorylation of these proteins was inhibited in the CPTAC cohort. Furthermore, we found that ECM receptors and the PI3K-Akt signaling pathway were overrepresented in downregulated proteins in EOC, with the phosphosites showing a downregulated pattern as well. Approved drug-target proteins were identified in the Drug–Gene Interaction database (DGIdb, https://dgidb.genome.wustl.edu/)[16] and were highlighted in the above pathways (Fig. 3a).

Clinical associations of dysregulated proteins in the above four pathways were assessed by univariate regression analysis. Multiple well-known therapeutic targets, such as CDK4, CDKN1B, and COL1A2, showed very high-risk scores for a mortality prognosis of EOC (Cox adj.$P$ values < 0.05, Fig. 3b and Supplementary Data 6). In addition,

ECM-receptor members including COL1A1, COL1A2 and COL6A3 were associated with RFS and showed significantly lower expression in tumor samples than in controls, with lower expression associated with longer survival rate (Cox adj.$P$ values < 0.05, Fig. 3b and Supplementary Data 6), which suggested that the receptors mediating the association between cellular and matrix components are altered in the EOC. Particularly, among these targetable proteins, CDK4, CDKN1B, and COL1A2 showed very high-risk scores in both OS and RFS of EOC (Cox adj.$P$ values < 0.05, Fig. 3b and Supplementary Data 6). To determine the stability of CDK4, CDKN1B, and COL1A2 protein expression, we used the Parallel Reaction Monitoring (PRM) analysis to target these proteins to quantify their expression levels and verified that these proteins were still under-expressed in EOC (Supplementary Fig. 2d). The above results suggested that targeting these proteins may prove beneficial in future clinical treatments.

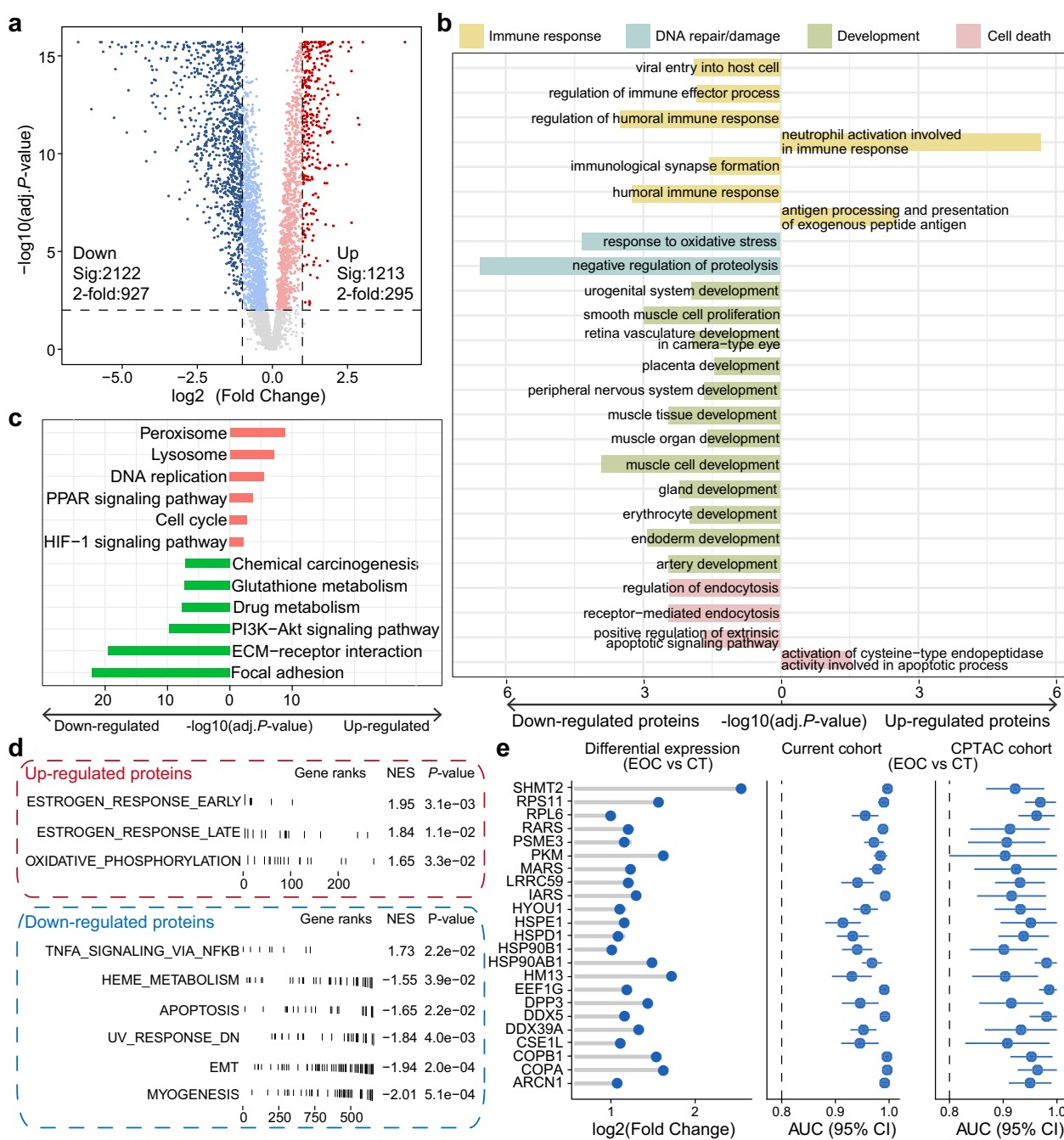

**Fig. 2 | Dysregulated proteins and pathways were identified by proteomic analysis. a** Volcano plot showed the dysregulated proteins upregulated or down-regulated in tumors. The adj.*P* values are calculated using two-sided Wilcoxon test (Benjamini & Hochberg (BH) adjusted *P* values). Pink and light blue colors represent proteins with adj.*P* values < 0.01, whereas dark red and blue represent proteins with adj.*P* values < 0.01 and |log2 (fold change)| >1. **b** Barplot colored by different categories show enriched biological processes (BPs) of dysregulated proteins. The adj.*P* values are calculated using hypergeometric test (BH adjusted *P* values). Left bars indicate BPs enriched in the downregulated proteins. Right bars indicate pathways enriched in the upregulated proteins. **c** Barplot of enriched KEGG pathways, with pink bars indicating pathways enriched in the upregulated proteins and green bars indicating pathways enriched in the downregulated proteins. The adj.*P* values are calculated using hypergeometric test (BH adjusted *P* values). **d** GSEA results show signatures evaluated in the context of marker sets representative for hallmarks of cancer. The statistical significance (nominal *P* value) of the enrichment score was calculated using an empirical phenotype-based permutation test procedure. **e** Lollipop plots show the fold-change value of proteins and forest plots show the area under curve (AUC) of these proteins. Current cohort: EOC samples (*n* = 239) and CT samples (n = 30). CPTAC cohort: EOC samples (*n* = 83) and CT samples (*n* = 20). The points and error bars show the AUC and 95% confidence interval (CI). Source data are provided as a Source Data file.

## Protein network characterization identifies candidate biomarkers and progression landmarks

To investigate the biological characteristics of our cohort from a proteome-wide perspective, we constructed a protein co-expression network based on protein expression profiling using weighted gene co-expression network analysis (WGCNA). A scale-free network was implemented with scale-free $R^2$ = 0.8 and soft-threshold power (β) = 4 as the soft-threshold values (Supplementary Fig. 3a). We performed the "cutreeDynamic" function with minModuleSize = 5, and identified up to 58 co-expression EOC modules (Supplementary

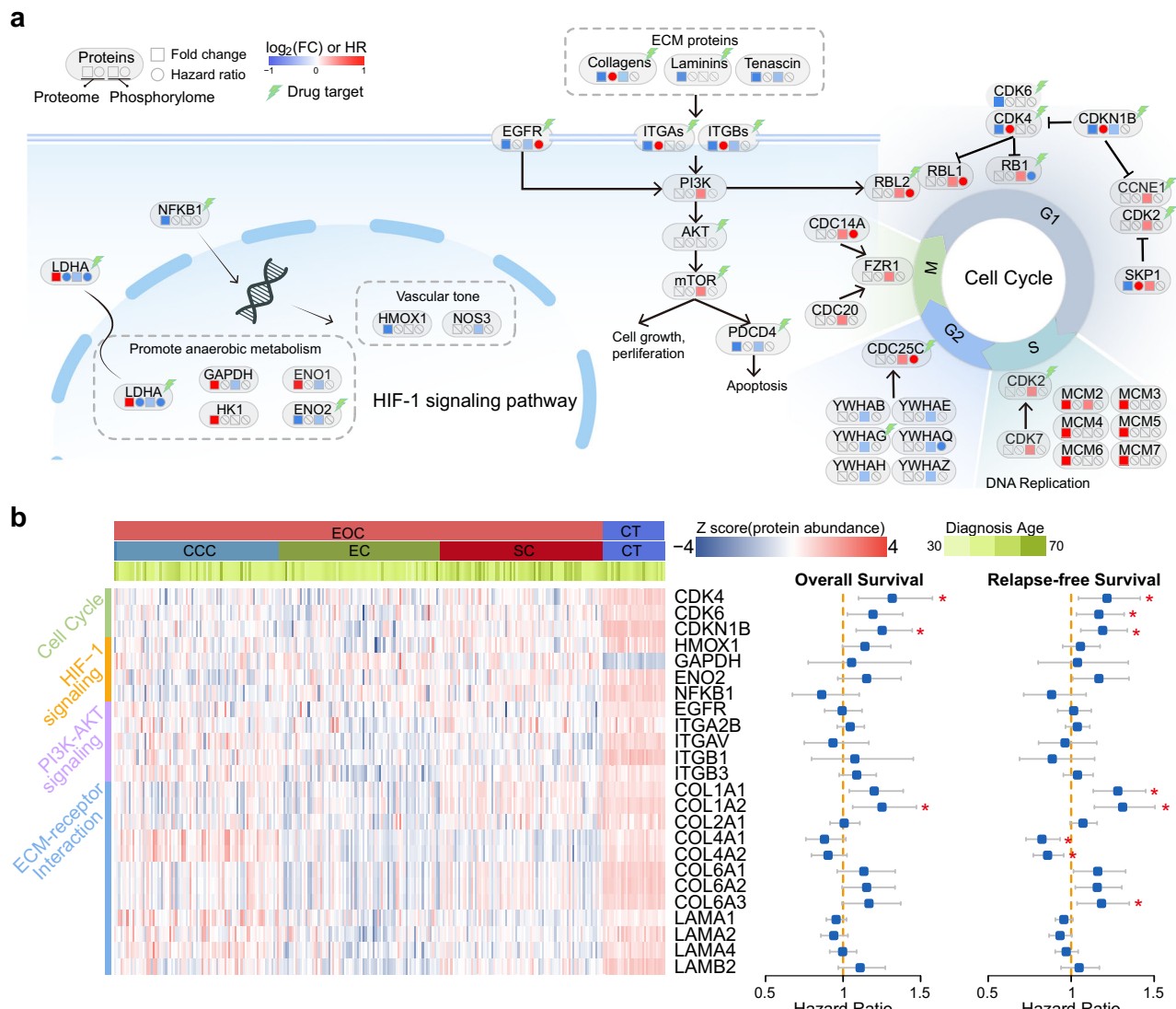

**Fig. 3 | Signaling pathway disturbances in EOC suggest therapeutic opportunities.** **a** Abnormal proteins in the KEGG pathway. **b** Heatmap showing the expression of proteins in four pathways, and forest plots showing HRs of proteins using Cox analysis (BH adjusted *P* values). The asterisks represent Cox adj.*P* values,

*adj.*P* values < 0.05 (adj.*P* values are shown in Supplementary Data 6). CCC samples (*n* = 80), EC samples (*n* = 79), SC samples (*n* = 80), and CT samples (*n* = 30). The points and error bars show the hazard ratio (HR) and 95% CI. Source data are provided as a Source Data file.

Fig. 3b). Among them, 41 modules were further extracted by exploring enriched BPs for each module by GO annotations (adj.*P* values < 0.05). Figure 4a illustrated the protein co-expression network of 896 nodes and 13,574 edges, which was annotated with the distinct biological functions of each module. For example, Module4, the largest co-expression module, was mainly participated in basic functions such as nucleic acid transport, RNA splicing and RNA localization. Module11 showed similar primary functions as Module4. Similarly, proteins in multiple modules were involved in DNA repair (Module9 and Module16), transport (Module27, Module45, etc.), and localization (Module14). As the second largest co-expressed module, Module1 was associated with a variety of important immune functions, such as acute inflammatory response, humoral immune response and immune effector process. In addition, Module18, Module36, and Module12 were also significantly enriched in immune-related BPs. In the co-expression network, 41.34% of the protein−protein interactions were known interactions in the STRING database (https://cn.string-db.org/)[17]. For the interactions in the co-expression network, the mean Pearson correlation was 0.643, and for the STRING database, it was 0.663 (Fig. 4b). In

addition, we examined the overlap between the above EOC modules and the histological subtype modules, where the protein associations between the modules were assessed using the hypergeometric Fisher's exact test (adj. *P* values < 0.05). The results showed that almost all histological subtype modules (97%) had a statistically significant overlap with the EOC modules (Supplementary Fig. 3c), demonstrating a better overlap between the EOC modules and the histological subtype modules.

Subsequently, we evaluated the abnormal expression levels of the proteins by comparing each histological subtype with CT separately (Supplementary Fig. 3d and Supplementary Data 7). When the relative protein abundances per pathogenic category were superimposed overlaid on the network, multiple modules (such as Module31, Module7, Module18, etc.) with differential regulation among the three histological subtypes were identified (Fig. 4c). The protein abnormalities and functional variations in the specific modules were considered essential. Interestingly, the proteins of Module31 (Fig. 4d) were not only differentially expressed among histological subtypes (Supplementary Fig. 4a), but also highly expressed in late stages (stage III and IV), elucidating the molecular dynamics of tumor

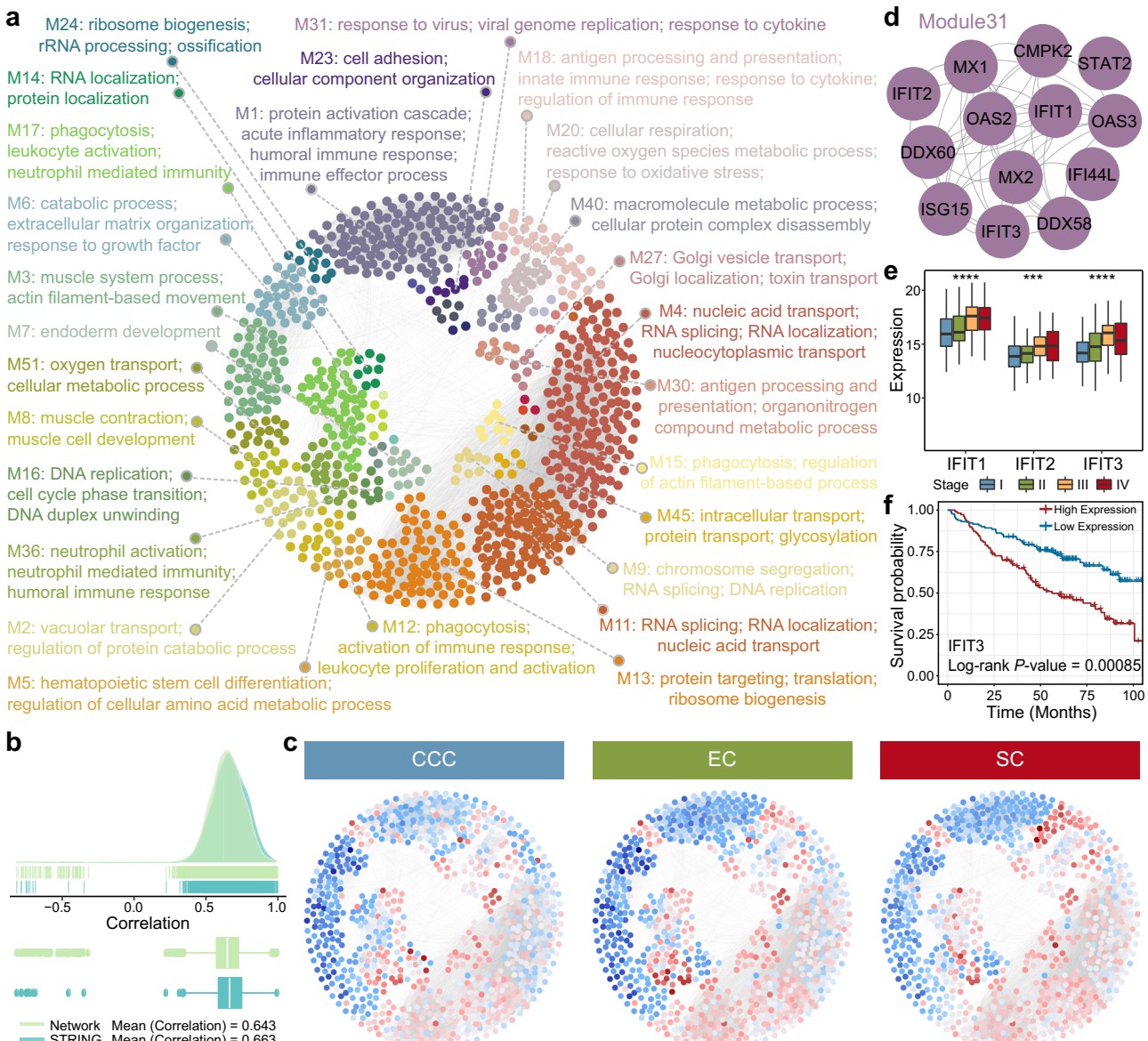

**Fig. 4 | Protein co-expression network and tumor progression landmarks.**
**a** Protein co-expression network of 896 nodes and 13,574 edges. Nodes are color-coded according to module membership. Representative enriched biological terms are shown for distinct modules. **b** Density plots of the pairwise protein–protein correlations for the interactions shown in the network. Number of interactions in Network ($n = 13,574$) and number of interactions in STRING ($n = 5611$). Boxplots show median (central line), upper and lower quartiles (box limits), min to max range. **c** Aberrant protein expression levels were superimposed on the network for each histological subtype. The red and blue dots represent up- and downregulated differentially expressed proteins, respectively. **d** Sub-network of Module31.
**e** Boxplots illustrated the abundances of IFIT1 ($P$ value = 3.3e-05), IFIT2 ($P$ value = 8.7e-04), and IFIT3 ($P$ value = 9.5e-07) in the different histological subtypes. The $P$ values are calculated using Kruskal–Wallis test. The asterisk character represents the significance of the expression discrepancy, ***$P$ value < 0.001 and ****$P$ value < 0.0001. Stage I samples ($n = 81$), Stage II samples ($n = 47$), Stage III samples ($n = 88$), and Stage IV samples ($n = 23$). Boxplots show median (central line), upper and lower quartiles (box limits), min to max range. **f** The Kaplan–Meier survival curve for IFIT3. Source data are provided as a Source Data file.

progression (Fig. 4e and Supplementary Fig. 4b). In particular, significant shorter OS time periods were evident in patients with higher expression of IFIT3 (Fig. 4f), indicating that IFIT3 could be a candidate biomarker and a progression landmark for EOC. Furthermore, ECM-receptor members including COL4A1, COL4A2, and LAMA1, which belong to Module7 (Supplementary Fig. 4c), were also associated with histological subtype and stage (Supplementary Fig. 4d, e). To verify the expression characteristics of these proteins, including IFIT1, IFIT2, IFIT3, COL4A1, COL4A2, and LAMA1, we performed the PRM analysis and found that the expression levels of these proteins in histological subtypes and stages were consistent with the above results, which reflected the reproducibility of the proteins (Supplementary Fig. 4f, g).

## Distinct molecular features and pathogenic mechanisms in histological subtypes

To gain insight into proteins that would be distinguished between the various subtypes, we applied the K-W test to identify differentially expressed proteins in the three histological subtypes. Next, a pairwise comparison between subtypes was performed. We assessed the overlap of differentially expressed proteins between SC with CCC and SC with EC as SC-specific proteins. We performed a similar analysis to identify the specific proteins of the two other subtype-specific proteins (Supplementary Fig. 5a). We identified 861 CCC-specific proteins, 423 EC-specific proteins, and 1094 SC-specific proteins (Fig. 5a). Next, we performed functional enrichment analysis on subtype-specific proteins, in which GO categories were grouped according to the

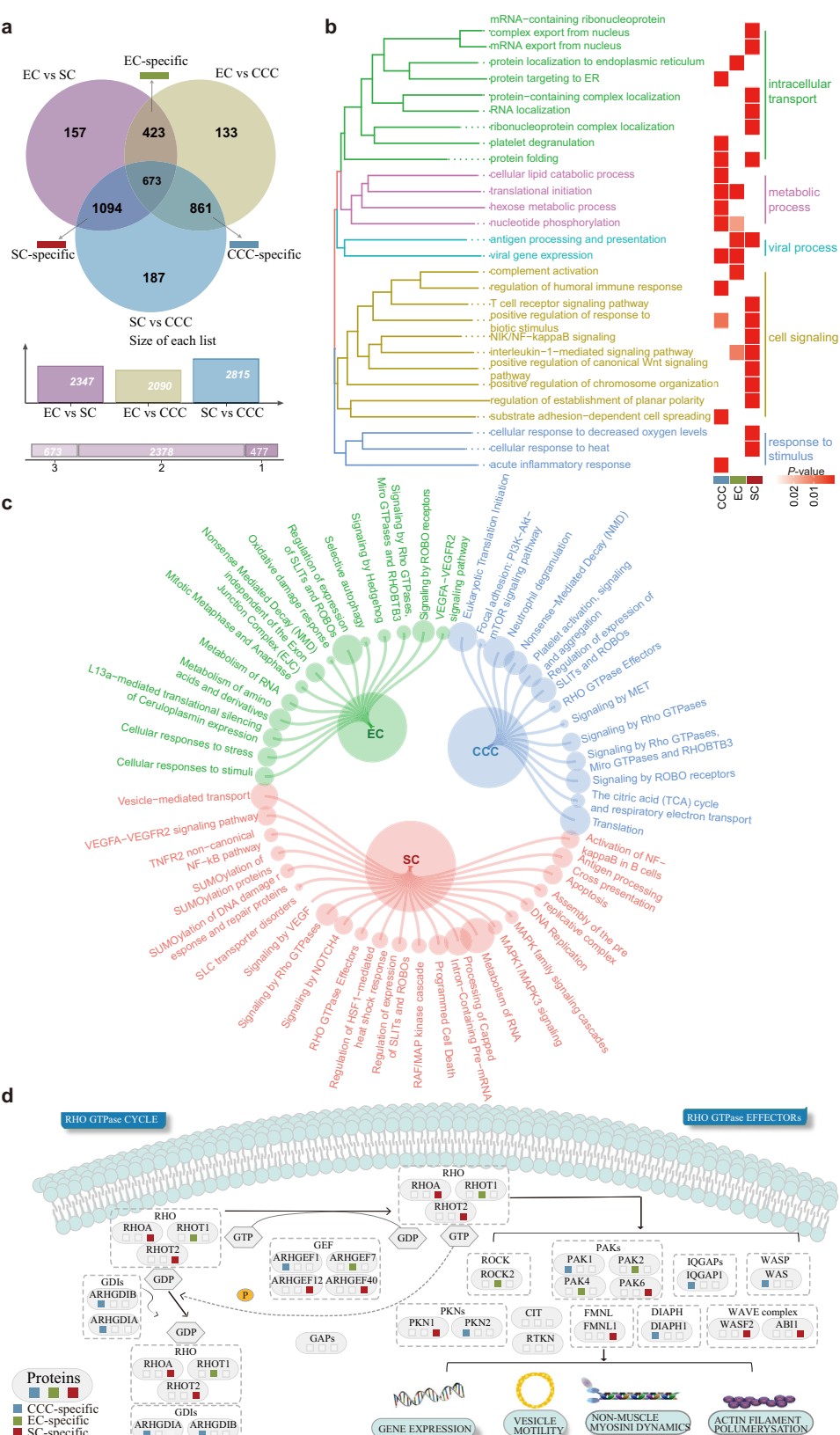

**Fig. 5 | Comparison and characterization of subtype-specific proteins. a** Venn diagram shows the overlap of differentially expressed proteins reported in different subtypes of EOC. **b**, **c** GO categories and molecular pathways enriched in differentially expressed subtype-specific proteins. CCC-specific proteins ($n = 861$), EC-specific proteins ($n = 423$), and SC-specific proteins ($n = 1094$). GO categories were grouped according to functional theme. The $P$ values are calculated using hypergeometric test. **d** Subtype-specific proteins were enriched in the pathway of Signaling by Rho GTPases. Source data are provided as a Source Data file.

functional theme. The CCC-specific proteins were enriched in metabolic processes such as cellular lipid catabolic process, translational initiation, and hexose metabolic process. The EC-specific proteins were enriched in viral process antigen processing and presentation. The SC-specific proteins were enriched in intracellular transport and cell communication such as protein-containing complex localization and substrate adhesion-dependent cell spreading (Fig. 5b). Next, we explored the biological significance of pathways using Metascape pathway analysis. The CCC-specific proteins were enriched in pathways such as PI3K-Akt signaling pathway, the citric acid (TCA) cycle and respiratory electron transport, and ECM-receptor interaction. The EC-specific proteins were enriched in pathways such as MHC class II antigen presentation, selective autophagy, and Oxidative damage response. The SC-specific proteins were enriched in pathways such as Programmed Cell Death, Activation of NF-κB in B cells and VEGFA-VEGFR2 signaling pathway (Fig. 5c). All three subtype-specific proteins were enriched in signaling pathways associated with Rho GTPases (Fig. 5d). The Rho family of small GTPases coordinate cell cycle progression, cell migration, and actin cytoskeleton dynamics[18]. A higher resistance of cultured metastatic EOC cells to chemotherapeutic drugs through the Rho/ROCK signaling pathway[19,20]. Furthermore, proteins in the pathway were differentially expressed in the three subtypes. For example, among the three subtypes, RHOA ($P$ values = 6.5e-13) and PAK1 ($P$ values = 6.3e-9) had the highest expression in SC and EC, respectively.

Next, we investigated common dysregulated proteins across the three subtypes and characterized the copy number variation of these proteins in TCGA (Supplementary Fig. 5b). We found no significant differences between the copy number variations of these dysregulated proteins. In addition, hallmark MTORC1 signaling was significantly associated with these common proteins (Supplementary Fig. 5c), and the common dysregulated proteins were enriched in protein autophosphorylation, axon development, cell junction, cell junction assembly and regulation of developmental growth (Supplementary Fig. 5d).

Based on the differences in tumor stages of three histological subtypes, the differences in protein and biological functions between early-stage (stages I and II) and late-stage (stages III and IV) of tumors were further investigated in each histological subtype separately. Supplementary Fig. 6a demonstrates the number of tumor progression landmarks for each histological subtype. Notably, there was a low overlap of tumor progression landmarks between histological subtypes, demonstrating their subtyping specificity. We also revealed specific KEGG signaling pathways that promote tumor progression for each histological subtype (Supplementary Fig. 6b). In particular, the SC subtype, with the worst survival rate, exhibited abnormalities in multiple signaling pathways, focusing on cellular processes (such as focal adhesion and regulation of actin cytoskeleton) and organismal systems (such as leukocyte transendothelial migration, FcγR-mediated phagocytosis, and chemokine signaling pathway) (Supplementary Fig. 6b and Supplementary Data 8). Among them, focal adhesion promotes tumor development and metastasis through effects on cancer cells and stromal cells of the tumor microenvironment[21]. The interaction of these aberrant signaling pathways may accelerate metastasis, recurrence, and even death in the SC population.

### Potential therapeutic targets for distinct histological subtypes
We tried to find potential drug targets for each histological subtype. First, the intersection of differentially expressed proteins in EOC and CT with subtype-specific proteins was used as a candidate protein list. Then, independent and significant prognostic proteins were identified for each histological subtype based on Kaplan–Meier curves and Cox regression analysis. In CCC, CSPG4 was highly expressed compared to the other two subtypes. Moreover, CSPG4 had prognostic value only in CCC and not in the other groups (Fig. 6a, b). Low expression of

TMEM87A was predominantly found in EC, and was significantly associated with good prognostic outcome in patients with EC (Fig. 6c, d). Protein MMP7 displayed excellent potential as a therapeutic target in SC, with subtype-specific high expression and prognostic ability (Fig. 6e, f). The prognostic value of MPP7 was also validated in the CPTAC dataset (log-rank $P$ values < 0.05).

Based on the prognostic value of MPP7 and the degree of malignancy of SC, we investigated the function of MPP7 in SC cells in vitro. Malignant behaviors including cell proliferation, cell migration, and cell invasion, cell cycle distribution, and cell apoptosis were assessed. Cell counting kit-8 (CCK-8) assays showed that shRNA-mediated MPP7 knockdown decreased cell viability (Supplementary Fig. 7a and Supplementary Data 9), indicating the inhibition of cell proliferation by MPP7 knockdown. Flow cytometric analysis of cell cycle demonstrated that MPP7 knockdown resulted in decreases in cells at the S phase and increases in cells at the G1 and G2 phase (Fig. 6g and Supplementary Data 10). The results implied that MPP7 was involved in the regulation of G1-S and S-G2 transition. Flow cytometric analysis of cell apoptosis elucidated that cell apoptosis was induced by MPP7 knockdown (Fig. 6h and Supplementary Data 11). Transwell assays suggested that cell migration and invasion were inhibited by MPP7 knockdown (Supplementary Fig. 7b, c and Supplementary Data 12 and 13). These findings uncovered that the biological function of MPP7 in SC. Taken together, we propose the further analysis of these subtype-specific proteins as promising therapeutic targets in the three histological subtypes.

## Discussion
Epidemiological studies have shown that OC is the second most common malignant gynecological tumor characterized by high mortality, increased recurrence rate, heterogeneity, and drug resistance[22]. Among them, EOC is the most common type of OC, accounting for more than 90% of cases, and it has the greatest fatality rate among malignant tumors of the female reproductive system[23]. Despite breakthroughs in chemotherapy and surgery for patients, this disease continues to represent a substantial threat to women, which is primarily due to the lack of symptoms in its early stages and apparent inter-tumor heterogeneity[24]. Thus, there is an urgent need to uncover the dysregulated molecular pathways and identify prognostic biomarkers and potential drug targets, which are essential for the prevention and treatment of EOC.

In clinical characteristics, we found that women in our cohort tended to be younger and had a more favorable outcome compared to the general EOC population, which is consistent with several previous studies[25–31]. In terms of age, previous evidence has shown that pregnancy history has been consistently related to OC risk[32]. Recent changes in fertility rates in China may have partly contributed to the early development of OC[33]. Another possible explanation for this disparity may be the younger age at diagnosis with better physical fitness and tolerance for aggressive treatment[26,28].

Although genetic analysis has made significant advances in precision oncology, it still has limitations since the functions encoded by the genome are predominantly executed at the protein level[34,35]. Proteomic-based analysis extends new perspectives for screening disease-associated biomarkers and candidate molecular targets, improving our understanding of malignancy transformation and therapeutic approaches[36–39]. In our study, we added another omics layer-based proteomic analysis of EOC, and provided complementary insights into this malignancy beyond our current genomic understanding. We performed a quantitative proteomic analysis of EOC and CT tissues and identified a series of differentially expressed proteins. Notably, most dysregulated proteins tended to be under-expressed in tumor tissues, which was consistent with a previous study[10]. For example, the expression abundance of multiple key proteins (such as COL1A1, COL1A2, ITGA2B, ITGB3, etc.) in the ECM-receptor interaction

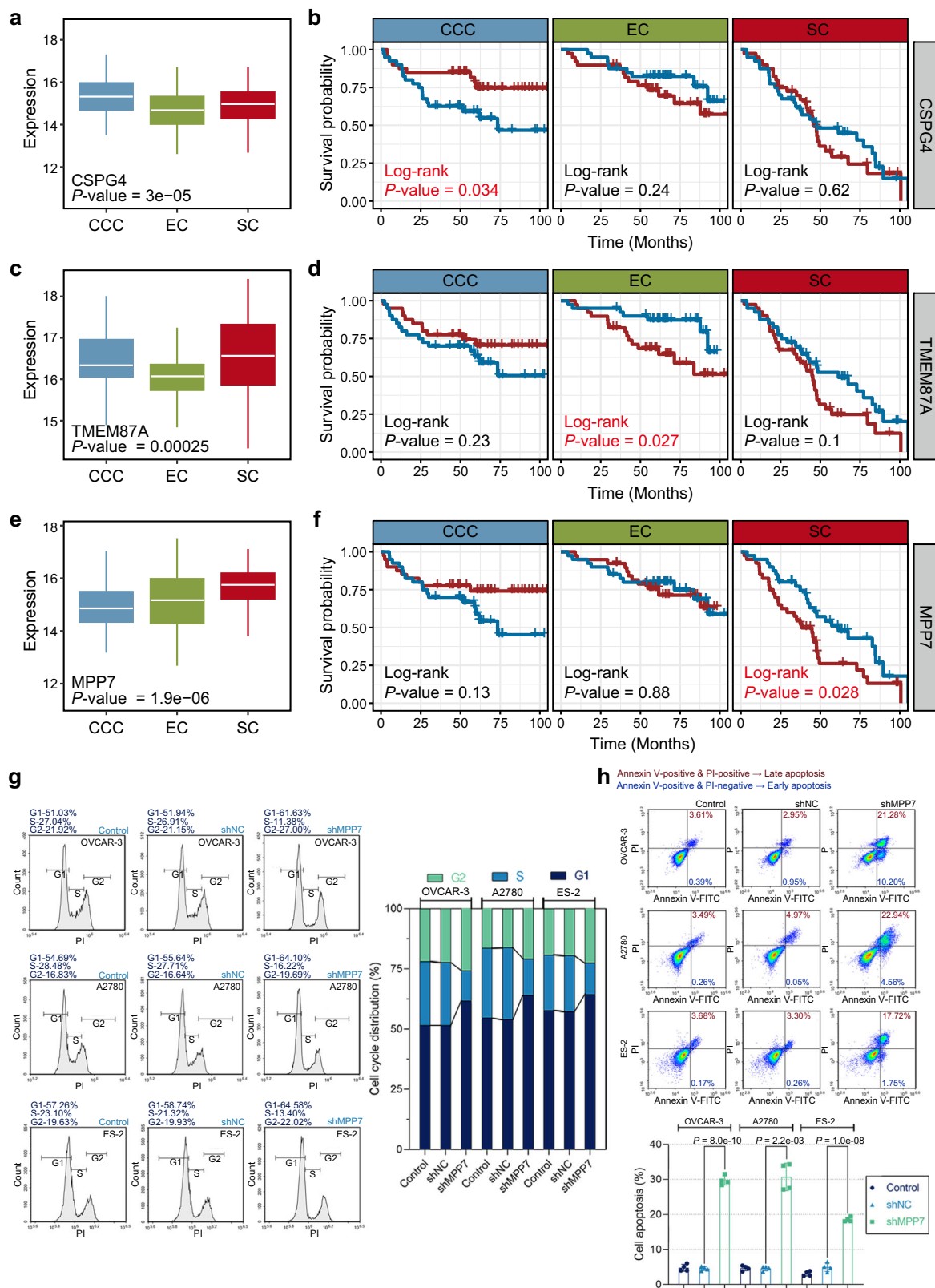

or PI3K-AKT signaling pathways were significantly downregulated[10]. And, significant enrichment of biological functions, such as humoral immune response, focal adhesion, and regulation of endocytosis, were corroborated[10]. Moreover, multiple disordered hallmarks of cancer were identified, including apoptosis, EMT and the TNFA signaling pathway, among others, suggesting oncogenic and invasive roles in EOC[40].

Particularly, disturbances in signaling pathways may suggest potential therapeutic opportunities. Uncontrolled cell proliferation is a hallmark of cancer[40]. The complex composed of cyclins and their associated cyclin-dependent kinases (CDKs) promotes cell cycle progression by phosphorylating and inactivating the retinoblastoma protein (RB)[41,42]. We observed that CDK4, CDK6, and CDKN1B participating in the cell cycle were all risk factors for the recurrence of

**Fig. 6 | Potential therapeutic targets for distinct histological subtypes. a, c, e** Expression levels of CSPG4, TMEM87A, and MPP7 among the respective histological subtypes. CCC samples ($n = 80$), EC samples ($n = 79$), and SC samples ($n = 80$). The $P$ values are calculated using the Kruskal–Wallis test. Boxplots show median (central line), upper and lower quartiles (box limits), min to max range. **b, d, f** Kaplan–Meier survival curves for patients expressing CSPG4, TMEM87A, and MPP7 in CCC, EC, and SC patients, respectively. CCC samples ($n = 80$), EC samples ($n = 79$), and SC samples ($n = 80$). Among them, the red line represents highly expressed protein and the blue line represents lowly expressed protein. **g** OVCAR-3, A2780, and ES-2 cells were infected with lentiviral vectors carrying shMPP7 or shNC.

Cell cycle distribution was analyzed by flow cytometry at 48 h after infection. **h** Cell apoptosis was measured by Annexin V-FITC/PI staining at 48 h after infection. The count of Annexin V-FITC-positive and PI-positive cells (Late apoptosis) and Annexin V-FITC-positive and PI-negative cells (Early apoptosis) was assessed by flow cytometry. Data are presented by mean ± standard deviation and analyzed by the one-way analysis of variance (ANOVA) followed by Tukey's tests or the Brown–Forsythe and Welch ANOVA tests followed by Tamhane T2 tests. The flow cytometry experiments were repeated four times. Source data are provided as a Source Data file.

EOC. Inhibition of CDKs blocks uncontrolled cell proliferation. The development of pharmacological inhibitors of CDKs has shown promising activity and clinical efficacy in the treatment of breast cancer[41,43–45]. There was also a phase II trial found that CDK4/6 inhibition with palbociclib was well tolerated and demonstrated single-agent activity in patients with recurrent ovarian cancer[46]. In addition, EOC cells spread through direct extension to the peritoneum, invade the underlying basement membrane and spread over the ECM to form metastatic implants, unlike most solid tumors that spread by lymphatic or hematogenous routes[47]. Collagen is the most abundant component in ECM proteins, which plays a critical role in cell proliferation, differentiation and maintenance of tissue homeostasis[48]. Among them, abnormal expression of COL4A1 and COL4A2 disrupts the strict regulation of the ECM and promotes the proliferation and invasion of cancer cells, which is often the main cause of cancer metastasis, recurrence and even death[49,50]. This evidence suggests that ECM proteins could be potential diagnostic markers for predicting EOC recurrence and promising drug targets for EOC treatment.

The construction of the protein co-expression network not only provides a comprehensive perspective of the biological features determining each histological subtype, but also rationalizes the selection of potential biomarkers and progression landmarks. Basic and immune functional modules dominate the network, and especially tend to more abundant protein expression in SC. Abnormalities in DNA repair and the immune system are developing fields of research in EOC pathogenesis[51,52]. For instance, cytotoxic T-cell infiltration has been shown to be associated with improved overall patient survival[53]. In particular, Module31 in our study not only participated in BPs related to antiviral and immune response, but also showed significant differences in histological subtypes and stages. In addition, IFIT proteins participate in the regulation of antiviral and immune response, apoptosis, cell population proliferation and migration[54–58]. Emerging evidence suggests that these IFIT proteins may also play roles in cancer progression[54]. For example, high IFIT3 expression predicted better response to IFN-α therapy in hepatocellular carcinoma patients[59], and its knockdown attenuated chemoresistance of pancreatic cancer cells to gemcitabine and paclitaxel therapy[60]. Intriguingly, IFIT3 characterized the molecular dynamics of tumor progression and predicted shortened OS of patients in our study, suggesting IFIT3 itself as a potential candidate biomarker and progression landmark for EOC.

There is considerable clinicopathologic heterogeneity in EOC[61], including a broad spectrum in histologic staging[62], and significant differences in mortality and recurrence rates[63], consistent with previous studies. Clearly, heterogeneity of EOC has a molecular basis. Examination of proteins with significantly different expression patterns among the three histological subtypes identified multiple subtype-specific proteins which were enriched in the Rho GTPase-related signaling pathways. Differences and commonalities in proteins from distinct histological subtypes may help us to better understand the mechanisms of tumor progression and facilitate the identification of potential therapeutic strategies. Given the large heterogeneity among histological subtypes, it is essential to consider heterogeneity

when considering potential therapeutic targets and treatment decisions. Based on the important role of MPP7 in SC at the levels of proliferation, cell cycle, apoptosis, migration, invasion, and prognosis, we considered it as a potential therapeutic target for SC. New et al. have analyzed MPP7 as positive regulator of pancreatic ductal adenocarcinoma cell survival and autophagy, providing a reasonable basis for considering this novel autophagy regulator as a therapeutic target[64]. These results may have significant implications for the development of histologic subtype-specific biomarkers and the coordination of therapeutic regimens.

We recognize several limitations. First, historical epidemiologic data have suggested that the incidence and survival rates of OC depend on ethnicity and geographical area[51,65]. Since East Asian backgrounds were significantly younger compared to other races and have an earlier stage of OC[30,66]. This could contribute to the fact that our cohort had favorable clinical outcomes than that of the general patients[25]. However, since our study only included patients from China, the data may not fully represent the entire population. This inevitably limits generalization to other populations and introduces the possibility of bias towards particular demographics[67]. Then, we integrated phosphoproteomic data from a previously published paper by CPTAC[10] to investigate the important role of proteins and their post-translational modifications in signaling pathways. Due to differences in patient race/ethnicity (the patients in our study were yellow, whereas the patients in CPTAC were predominantly white), this may have imposed some limitations on the use of phosphoproteomic data. Further research is necessary to investigate and confirm the findings reported hereof to be clinically meaningful. In addition, we only obtained mass spectrometry-based proteomic data, lacking data on the transcriptome, mutations, and copy number variation data. This limits our study to proteomic, and multi-omics data will be used to deepen the study of EOC histological subtypes in the future.

In conclusion, we have provided a comprehensive characterization of EOC based on proteomic analysis, broadening our understanding of the molecular features associated with this lethal malignancy. Proteomic characterization of EOC and CT tissue samples has revealed features of protein dysregulation and disruption of key signaling pathways. The protein network reconstitution not only provides a global perspective of the biological features of each histological subtype, but also suggests potential prognostic biomarkers and progression landmarks. The marked histological subtype specificity of EOC is manifested in various aspects, which include staging, survival rate, recurrence rate and regulation of signaling pathways. Based on the distinct differences in protein abundance, prognostic ability, and drug ability of histological subtypes, we also predict potential targets for each subtype separately. The in vitro experiments further highlighted the importance of MPP7 in the malignant behavior of SC cells. Although experimental verification of the biological functions of MPP7 in SC cells may not represent the prognostic value of MPP7, our study provides a potential target for drug therapy in SC that needs to be further validated in future research. This comprehensive analysis has thus provided broad insights into the biological characterization, clinical diagnosis and therapeutics of EOC.

## Methods

### Clinical specimens acquisition

All clinical samples, as well as the corresponding clinical information, were collected after approval by the Institutional Review Board of Shengjing Hospital of China Medical University (2020PS265K) with written and informed consent from the patients who were not compensated. A total of 239 formalin-fixed, paraffin-embedded (FFPE) ovarian epithelial tissues were acquired from newly diagnosed EOC patients undergoing primary debulking surgery at the Shengjing Hospital of China Medical University (Shenyang, China) from 2013 to 2019. Detailed information on sample collection, evaluation, and processing is in Supplementary Method 1. All patients were treated with both carboplatin and paclitaxel. The 239 patients were included in the present analysis with three histological types: SC samples ($n = 80$), EC samples ($n = 79$) and CCC samples ($n = 80$). Based on the characteristics of EOC, it is difficult to obtain para-carcinoma tissue[68]. Therefore, the histologically normal ovarian tissues ($n = 30$) taken from cases of uterine fibroids were used as CT samples, in which the ovary was surgically removed incidental to radical surgery. The median age of CT samples at time of surgery was 55 years old and ranged between 41 and 71 years old. All of these samples were examined by two experienced pathologists who confirmed the diagnosis of the disease samples. Eligible tumor samples contain at least 90% tumor cells.

Tumor stages were calculated according to the International Federation of Gynecology and Obstetrics (FIGO) criteria and histologic typing system of the WHO, respectively[69]. OS was determined as the interval between surgery and death from any cause or the date of last follow-up (December 31, 2021) for patients who were still alive. The primary endpoint was RFS, defined as the time from completion of primary surgery to the first progression or recurrence of disease or death from any cause[70].

### FFPE ovarian tissue section preparation

Detailed information on sample collection, evaluation, and processing has been provided below.

**Obtaining a fresh specimen.** Cut small blocks of tissue $1\,cm^2 \times 0.4\,cm$, and place them in a histological/tissue processing cassette. Cautious: (1) The ovarian tissue was removed gently to avoid trauma by an expert gynecologist; (2) Specimen is not allowed to dry out prior to fixation; (3) Avoid contaminating fresh specimens with foreign chemicals or substances such as disinfectants; (4) Each specimen should be properly identified and name, pathology number, and other details recorded as soon as possible; (5) Fixation is always carried out promptly. If it is necessary that a specimen remains unfixed for a short period of time, it should be refrigerated at 4 °C.

**Fixation.** To the tissue, add 20× the tissue volume 10% neutral formalin. Cautious: (1) The specimen is placed in formalin, this will slowly penetrate the tissue causing chemical and physical changes that will harden and preserve the tissue and protect it against subsequent processing steps; (2) An adequate volume of fixative (ratio of at least 20:1) is used in a container of an appropriate size. This avoids distortion of the fresh specimen and ensures good quality fixation; (3) Ideally, specimens should remain in fixative for long enough for the fixative to penetrate every part of the tissue and then for an additional period to allow the chemical reactions of fixation to reach equilibrium (fixation time). Generally, this will mean that the specimen should fix for between 6 and 48 h.

**Dehydration.** Because melted paraffin wax is hydrophobic (immiscible with water), most of the water in a specimen must be removed before it can be infiltrated with wax, a typical dehydration sequence for specimens not more than 4-mm thick would be: (1) 80% ethanol 1 h; (2) 90% ethanol 1 h; (3) 95% ethanol 1 h; (4) 95% ethanol 1 h; (5) 100% ethanol

1 h; (6) 100% ethanol 1 h; (7) 100% ethanol 1 h. Cautious: Processing reagents are replaced strictly according to established guidelines.

**Clearing.** A popular clearing agent is xylene, and multiple changes are required to completely displace ethanol, a typical clearing sequence for specimens not more than 4-mm thick would be: (1) xylene 1 h; (2) xylene 1 h. Cautious: Processing reagents are replaced strictly according to established guidelines.

**Wax infiltration.** A typical infiltration sequence for specimens not more than 4-mm thick would be: (1) wax 1 h; (2) wax 1 h; (3) wax 1 h. Cautious: High-quality wax is used for infiltration to ensure high-quality blocks that are easy to cut.

**Embedding.** This step is carried out using an "embedding center" where a mold is filled with molten wax and the specimen is placed into it. The specimen is very carefully oriented in the mold because its placement will determine the "plane of section", an important consideration in both diagnostic and research histology. A cassette is placed on top of the mold, topped up with more wax, and the whole thing is placed on a cold plate to solidify. When this is completed, the block with its attached cassette can be removed from the mold and is ready for microtomy. It should be noted that, if tissue processing is properly carried out, the wax blocks containing the tissue specimens are very stable and represent an important source of archival material. Cautious: (1) Specimens are carefully oriented, competent grossing ensures flat surfaces on most specimens; (2) A mold of suitable size is always chosen for each specimen; (3) Specimens are handled gently during embedding; (4) Before handling tissue, forceps are heated to the point where the wax just melts; (5) Before handling tissue, forceps are heated to the point where the wax just melts; (6) Molds are filled to an optimum level and do not overflow.

### Protein profiling and quality control

**Materials and reagents.** The materials and reagents required for sample preparation are listed in Supplementary Data 14.

**Protein extraction.** After dewaxing, the samples were transferred to 1.5-mL centrifuge tubes, and four times the volume of cracking buffer (1% SDS, 1% protease inhibitor) was added for ultrasonic cracking. After centrifugation for 10 min at $12,000 \times g$ and 4 °C, cell fragments were removed, and the supernatants were transferred to new centrifuge tubes. Protein concentration was determined using a BCA kit.

**Trypsin digestion.** The protein from each sample was enzymatically hydrolyzed in equal quantities, and the volume was adjusted to be consistent with the lysate. One time the volume of pre-cooled acetone was added, followed by vortex mixing, then four times the volume of pre-cooled acetone was added and precipitation was performed for 2 h at −20 °C. After centrifugation at $4500 \times g$ for 5 min, the supernatant was discarded, and the precipitate was washed with pre-cooled acetone two times. After drying the precipitate, triethylammonium bicarbonate with a final concentration of 200 mM was added and the precipitate was broken up by ultrasound, then trypsin was added at a ratio of 1:50 (protease: protein, m/m) and the sample was enzymolized overnight. Dithiothreitol was added at a final concentration 5 mM and the sample was reduced for 60 min at 37 °C. Finally, iodoacetamide was added at a final concentration of 11 mM, and the sample was incubated for 45 min at room temperature in the dark.

**LC-MS/MS analysis.** The peptides were dissolved by liquid chromatographic mobile phase A and separated by the NanoElute ultra-high performance liquid system. Mobile phase A was an aqueous solution containing 0.1% formic acid and 2% acetonitrile. Mobile phase B contained 0.1% formic acid and 100% acetonitrile solution. Liquid phase

gradient settings were 0–60 min, 5–22% B; 60–69 min, 22–31% B; 69–72 min, 31–80% B; 72–75 min, 80% B; the flow rate was maintained at 400 Nl/min. The peptides were separated by the ultra-performance liquid phase system and then injected into a capillary ion source for ionization and analyzed by a timsTOF Pro mass spectrometer. The ion source voltage was set at 1.75 kV, and the peptide parent ions and their secondary fragments were detected and analyzed using high-resolution TOF. The scanning range of the secondary mass spectrometry was set to 100–1700. Data acquisition was performed in parallel cumulative serial fragmentation (PASEF) mode. After first-order mass spectrometry collection, the secondary spectra with the charge number of parent ions in the range of 0–5 were collected in PASEF mode ten times. The dynamic exclusion time of tandem mass spectrometry scanning was set to 30 s to avoid repeated scanning of parent ions.

**Database search.** The resulting MS/MS data were processed using the Maxquant search engine (v1.6.15.0)[71,72]. Tandem mass spectra were searched against Homo_sapiens_9606_SP_20210721.fasta. Trypsin/P was specified as the cleavage enzyme allowing up to two missing cleavages. The mass tolerance for precursor ions was set at 20 ppm in the first search and Main search. Carbamidomethyl on Cys was specified as fixed modification and oxidation on Met was specified as variable modifications. The false discovery rate was adjusted to <1%.

## Proteomic data filtering and normalization
Label-free quantitation (LFQ) intensity of 269 samples (239 EOC and 30 CT samples) were obtained from the Maxquant result files. Proteins with missing values in more than half of the samples were removed[8]. As a result, 4447 proteins out of a total of 8257 proteins were retained. The LFQ intensity of the 4447 proteins was normalized using the normalized quantile functions in the R package "limma"[73]. Missing values were imputed using the DreamAI algorithm[74].

## Differential expression analysis of the proteome
Differential proteomic analysis was conducted on 4447 quantifiable proteins in a total of 8257 proteins detected. We performed Wilcoxon tests to identify the dysregulated proteins with a statistically significant $P$ value between EOC and CT patients. The $P$ values were corrected by the Benjamini & Hochberg (BH) procedure. Significantly up- or downregulated proteins were extracted by a threshold of adj.$P$ value < 0.01 and |log2 (fold change)| >1. We also used "limma" analysis to adjust for clinicopathological characteristics and calculate protein abundance differences. In addition, we compared the differentially expressed proteins between histological subtype and CT separately using the above methods.

To assess differentially expressed proteins across treatment groups, the Kruskal–Wallis test (K-W test) was used to identify differentially expressed proteins among the three histological subtypes of EOC (SC, EC, and CCC). Post hoc tests were performed to identify the differentially expressed proteins between any two subtypes (adj.$P$ values < 0.05, R package "PMCMRplus").

## Survival analysis
For the clinicopathological analysis, Fisher's exact test (two-sided) was performed. Kaplan–Meier curves and log-rank tests were used to compare OS or RFS among the proteomic subtypes. Clinical associations of protein expression were examined using the Cox proportional hazards model, and $P$ values were adjusted using the BH procedure. Univariable and multivariable Cox regression analysis were used to estimate the hazard ratios (HR), 95% confidence intervals (CI), Cox $P$ values, and Cox adj.$P$ values of each protein.

## Functional enrichment analysis
Comprehensive function annotation of proteins, including GO[75,76], KEGG[77] and GSEA[78], was performed on R package "clusterProfiler"[79],

Metascape (http://www.metascape.org/) and R package "fgsea"[80] to identify BPs, KEGG pathways, Reactome gene sets[81], WikiPathways[82], and Hallmark gene sets[78,83] in which dysregulated proteins were enriched. The adj.$P$ values < 0.05 were considered statistically significant. R packages "simplifyEnrichment"[84] and "GOSemSim"[85] were used to cluster GO terms based on similarity matrices of functional terms. The function "simplify" of R package "clusterProfiler" was used to remove redundancies of enriched GO results.

## Weighted correlation network analysis
The whole quantifiable proteins were identified by WGCNA to construct a protein co-expression network using R package "WGCNA"[86] to generate the co-expression network and modules. Scale-free $R^2 = 0.8$ was set in order to make the network more consistent with scale-free characteristics. Meanwhile, the adjacency matrix was transformed into a topological overlap matrix (TOM) to reduce noise and spurious correlation. Network construction and module identification were performed based on (TOM) similarity. Other parameters were set as follows: soft-threshold power (β) = 4, "cutreeDynamic" function, minModuleSize = 5. The co-expression network was visualized using Cytoscape v3.6.0[87].

## Parallel reaction monitoring
PRM is an ion monitoring technology based on high-resolution, high-precision mass spectrometry, which can selectively detect target proteins, so as to achieve quantification of target proteins. Detailed information about the PRM analysis is provided below.

**Trypsin digestion.** The protein sample was added with 1 volume of pre-cooled acetone, vortexed to mix, and added with 4 volumes of pre-cooled acetone, precipitated at −20 °C for 2 h. The protein sample was then redissolved in 200 mM TEAB and ultrasonically dispersed. Trypsin was added at 1:50 trypsin-to-protein mass ratio for the first digestion overnight. The sample was reduced with 5 mM dithiothreitol for 60 min at 37 °C and alkylated with 11 mM iodoacetamide for 45 min at room temperature in darkness. Finally, the peptides were desalted by Strata X SPE column.

**LC-MS/MS analysis.** The tryptic peptides were dissolved in 0.1% formic acid (solvent A), directly loaded onto a homemade reversed-phase analytical column. The gradient was comprised of an increase from 6% to 20% solvent B (0.1% formic acid in 98% acetonitrile) over 16 min, 20% to 30% in 6 min and climbing to 80% in 4 min then holding at 80% for the last 4 min, all at a constant flow rate of 500 nL/min on an EASY-nLC 1000 UPLC system. The peptides were subjected to NSI source followed by tandem mass spectrometry (MS/MS) in Q ExactiveTM Plus (Thermo) coupled online to the UPLC. The electrospray voltage applied was 2.1 kV. The $m/z$ scan range was 390 to 1135 for full scan, and intact peptides were detected in the Orbitrap at a resolution of 70,000. Peptides were then selected for MS/MS using NCE setting as 27 and the fragments were detected in the Orbitrap at a resolution of 17,500. A data-independent procedure that alternated between one MS scan followed by 20 MS/MS scans. Automatic gain control was set at 3E6 for full MS and 1E5 for MS/MS. The maxumum IT was set at 210 ms for full MS and auto for MS/MS. The isolation window for MS/MS was set at 1.6 $m/z$.

**Data analysis.** The resulting MS data were processed using Skyline (v.21.1). Peptide settings: enzyme was set as Trypsin [KR/P], Max missed cleavage set as 0. The peptide length was set as 7-25, Variable modification was set as Carbamidomethyl on Cys and oxidation on Met, and max variable modifications were set as 3. Transition settings: precursor charges were set as 2, 3, ion charges were set as 1, ion types were set as b, y. The product ions were set as from ion 3 to the last ion, the ion match tolerance was set as 0.02 Da.

## Experimental methods and statistical analysis

**Cell culture and infection.** Ovarian cancer cell lines OVCAR-3, A2780, and ES-2 were purchased from icellbioscience (China) and cultured in RPMI-1640 medium (Solarbio, China) containing 20% FBS (TIANHANG, China), RPMI-1640 medium containing 10% FBS, and McCOY's 5 A medium (Procel, China) containing 10% FBS at 37 °C with 5% $CO_2$, respectively. All human cancer cell lines were authenticated by STR profiling and mycoplasma-negative. Cells were infected with lentiviral vectors carrying short hairpin RNA (shRNA) targeting MPP7 (shMPP7) or its negative control shRNA (shNC) (OVCAR-3: MOI = 30; A2780: MOI = 10; ES-2: MOI = 5).

**CCK-8 assay.** Cells ($3 \times 10^3$) were seeded onto a 96-well plate and harvested at 0, 24, 48, 72, or 96 h after infection. The CCK-8 assays were performed using the CCK-8 assay kit (Wanleibio, China) according to the manufacturer's instructions. The optical density (OD) was detected at 450 nm using a BioTek 800 TS plate reader (Agilent, USA).

**Flow cytometry.** Cell cycle distribution and cell apoptosis were analyzed using flow cytometry. For cell cycle analysis, cells were fixed with 70% cold ethanol and washed with PBS at 48 h after infection. After incubation with RNase A at 37 °C for 30 min, the cells were stained with Propidium Iodide (PI) for 30 min. For cell apoptosis detection, cells were washed with PBS at 48 h after infection and resuspended in binding buffer. Then, the cells were stained with Annexin V-FITC and PI for 15 min. The Cell Cycle Analysis Kit and the Annexin-FITC-PI Staining Kit were purchased from Wanleibio (China). The signal of PI and Annexin V-FITC was detected by a NovoCyte flow cytometer (Agilent, USA). The results obtained from flow cytometric analysis were analyzed by the NovoExpress software (version 1.4.1, Agilent, USA). Gating strategies for flow cytometric analysis are shown in Supplementary Fig. 8.

**Transwell assay.** The transwell chambers used for migration assays or the matrigel-coated transwell chambers used for invasion assays were placed into 24-well plates. Cells were resuspended in serum-free medium at 48 h after infection and seeded in the transwell upper chamber at 5000 (Migration assay) or 50,000 (Invasion assay) cells per well. After 24 h, the cells on the lower side of the transwell membrane were fixed with 4% PFA (Aladdin) for 20 min, stained with 0.5% crystal violet (Amresco, USA) for 5 min, and counted under a microscope (OLYMPUS, Japan).

**Statistical analysis.** GraphPad Prism 8 (GraphPad Software, USA) was used for statistical analyses. The Shapiro-Wilk normality test and the Brown–Forsythe test were used for the analysis of normal distribution and variance homogeneity, respectively. Data, which were normally distributed and had equal variances, were analyzed using one-way or two-way analysis of variance (ANOVA) followed by Tukey's tests. Data, which were normally distributed and had unequal variances, were analyzed using Brown–Forsythe and Welch ANOVA tests followed by Tamhane T2 tests. Data are presented as mean ± standard deviation (SD). A $P$ value < 0.05 was considered statistically significant.

## Reporting summary

Further information on research design is available in the Nature Portfolio Reporting Summary linked to this article.

## Data availability

All relevant data supporting the key findings of this study are either downloaded from open repositories or have been uploaded to such repositories and are publicly available. The mass spectrometry proteomic data generated in this study have been deposited in the ProteomeXchange Consortium via the PRIDE partner repository under accession code PXD033741. The mass spectrometry proteomic data generated in this study have also been deposited in the OMIX under accession code OMIX002719, in accordance with the necessary approval from the Chinese Ministry of Science and Technology related to export the genetic information and materials associated with this study. The exosome protein lists used in this study are available in the ExoCarta (http://www.exocarta.org/) and Vesiclepedia (http://microvesicles.org/) databases. The protein–protein interactions used in this study are available in the STRING (https://cn.string-db.org/) database. The approved drug-target protein lists used in this study are available in the Drug-Gene Interaction database (DGIdb, https://dgidb.genome.wustl.edu/). The remaining data are available within the Article, Supplementary Information or Source Data file. Source data are provided with this paper.

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

## Acknowledgements

We want to thank Xin-Rui Meng (Jingjie PTM BioLab Co. Ltd, China), Dr. Zheng-Wei Yuan (Shengjing hospital of China Medical University), and Chao Wang (Wanlei Biotechnology Co. Ltd, China) for the help of the data interpretation. This work was supported by the National Key R&D Program of China (No. 2022YFC2704205 to Q.W.), National Natural Science Foundation of China (No. 82073647 and No. 82373674 to Q.W., No.82103914 to T.G. and No.32070672 to S.N.), LiaoNing Revitalization Talents Program (No. XLYC1907102 to Q.W.), Outstanding Scientific Fund of Shengjing Hospital (No. M1150 to Q.W.), Clinical Research Cultivation Project of Shengjing hospital (Song.G.), and 345 Talent Project of Shengjing Hospital of China Medical University (T.G. and Q.W.).

## Author contributions

Q.J.W. and S.W.N. conceived the original ideas and designed the project. F.H.L., Y.L.H., M.Z., S.Y., Q.X., X.Y.L., and Y.Z.L. performed sample collection, follow-up. H.L.X., Y.K., Song Gao, X.Q., D.H.H., Y.Y.Z., and Y.H.Z. supervised the pathology data analysis and interpretation. X.P. and X.Y.W. performed sample processing, protein extraction, mass spectrometry analysis, and data upload. Shuang Guo, H.X.Z., X.P., X.X.Z., Y.L.W., X.X.M., and X.Y.W. performed the data processing, bioinformatics analysis and drafting. Q.J.W., S.W.N., T.T.G., and Shuang Guo wrote the manuscript. All authors reviewed and approved the final version of the manuscript.

## Competing interests

The authors declare no competing interests.
