## [Peer Review File · Nature Communications]

Proteomic characterization of epithelial ovarian cancer delineates molecular signatures and therapeutic targets in distinct histological subtypesReviewers' Comments:

Reviewer #1:

Remarks to the Author:

Summary:

The paper entitled Proteogenomics Characterization of Epithelial Ovarian Cancer Delineates Molecular Signatures and Therapeutic Targets in Distinct Pathological Subtypes by Gong et al. aims to profile the proteomic features of Epithelial Ovarian Cancer with different histological stages and to integrate this information with patients' prognosis. The focus of the authors was on identifying proteomic signatures that associates with patients' survival. Overall, the paper provides the resource to the proteomics and EOC communities and fits the scope of Nature Communications, however there are some concerns need to be satisfactorily addressed.

Major points:

1. The title of the paper needs to be changed, since the author did not perform genomic analysis.
2. It was not fully clear how these samples are actually collected, evaluated and processed to ensure are certain degree of tumor cellularity?
3. Also how did the NAT samples were chosen, what criteria did the author apply?
4. The paper lack of details on the quantification of proteomic data. It is unclear how many proteins were detected in each sample. Also, did the author utilize imputation strategy for missing values?
5. Moreover, the author integrated phosphoproteomic data from previously published paper by CPTAC, while direct comparison/integration to the CPTAC cohort may be challenging due to the different proteomics technologies used in both studies. Are the baseline characteristics of patients in the current cohort different from CPTAC cohort? Did the difference impact patients' prognosis and further analysis?
6. The comparison to other published data sets is limited. This is important because this study is based on FFPE archival samples and not fresh frozen tissues. Please comment? Also, how comprehensive and correct is the presentation of proteomics based on FFPE. It would be important to validate some of the findings in this study rather than adding new omics data.
7. Functional experiments should be done, since the current study is based on bioinformatic analysis, it is important for the author to perform functional analysis utilizing PDCs and PDX models.
8. Also, the research lacks of validation either at functional experiment level or at cohort level. It is interesting that the author conducted PRM analysis to verify proteomic biomarkers, however, the accuracy and applicable of those proteins need to be further confirmed in a independent validation cohort.
9. Both raw data and processed data from transcriptomic and proteomic analysis should made available in public repositories such as Scientific Data.
10. All statistical analysis should be checked by statisticians.

Reviewer #2:

Remarks to the Author:

This paper comprehensively analyzes novel insights into the 580 biological characterization for the improvement of clinical diagnosis and therapeutics of EOC. It is my opinion that this will contribute significantly to this field due to its novel and thorough approach. Authors provide sufficient evidence to support their claims. Methodology is sound and meets the criteria of the standards in our field. Figures and legends are well done and discussion underscores importance of their findings.

Minor critique: the figures 4 and 5 have very small sized font, please consider increasing font or rearranging figure so as not to be so filled with words (can be overwhelming and hard to follow).

Reviewer #3:

Remarks to the Author:

Proteogenomics Characterization of Epithelial Ovarian Cancer Delineates Molecular Signatures and Therapeutic Targets in Distinct Pathological Subtypes by Gong et al aims to characterize the proteomic landscape of different subtypes of ovarian cancer and the diagnostic and prognostic value of these proteins.

The authors used 239 cancer samples and 30 normal control ovarian tissue to characterize the different proteomes and to discover aberrant pathways. The main strength of this paper is the large sample size, the use of negative control ovarian samples and the comprehensive proteomics analysis. In addition, the paper identifies several pathways which could be potentially used as future targets in ovarian cancer.

Questions to the authors:

-In the methods section, the authors state that a total of 239 formalin-fixed, paraffin-embedded (FFPE) ovarian epithelial tissues were acquired from newly diagnosed EOC patients undergoing primary debulking surgery at the Shengjing Hospital of China Medical University (Shenyang, China) from 2013 to 2019. The last survival follow-up data on these patients were in March 2021. However, based on the Supplementary figure – half of the serous cancer patients were still alive, and only about 40% of the endometrioid and clear cell patients had recurred. This seems to be a much more favorable outcome than one would be expected in a general OC population. How do the authors explain this difference?

-As much of the proteomics analysis is based on RFS/OS analysis, less than half of the cohort progressed, and 75% of the cohort is still alive. Do the authors have more updated survival data on this cohort of patients?

-The authors comment on the heterogeneity of OC, although in their study, they mainly compare different subtypes to each other. Therefore, it is not surprising that the different subtypes have a distinct proteomic landscape. Did the authors look at the difference in the same subtype in early-stage disease (stages 1 and 2) vs. late-stage (stages 3&4)?

-In Supplementary Table 1 – please define what “Age” means here – is this the mean or median age of the patients? And SD?

-The median age of ovarian cancer, in general, is 63 – while in this cohort of patients is 52, which is 11 years younger than the expected age. Although clear cell OC, on average a few years younger, this difference is also seen in the serous cohort. Could you please explain what could be driving the ovarian cancer diagnosis at such a young age in this cohort of patients?

-Please provide survival data in months rather than days. What do these “time” numbers mean in the table OS and RFS? As in almost all the cohorts, less than half of the patients recurred/died – this could not mean median RSF/OS.

-Do the authors have information about genetic testing on these patients?

-Could you please provide comments on subsequent treatments in recurrent disease

-Please comment on the control ovarian samples – what was the age of those patients, and how were these specimens obtained?

-Please provide data on race in the manuscript and if most of the patients are of East Asian descent – please acknowledge this in the discussion section as a limitation. East Asian patients do have more favorable clinical outcomes compared to other races – and higher rates of clear cell OC. Thus, some of these findings may not be completely generalizable to a different cohort of patients.

Reviewer #4:

Remarks to the Author:

Gong et al. quantified the proteome of 269 epithelial ovarian cancer (EOC) samples. The protein ranking lists were generated from a differently expressed test between EOC and normal adjacent tissues, and between EOC histological subtypes. Functional enrichment analysis and protein network analysis were also conducted to interpret the proteome data. It's an opportunity to generate molecular profiles to gain biological and clinical insights into EOC in addition to the well-known Clinical Proteomic Tumor Analysis Consortium (CPTAC) discoveries. However, there is not enough molecular data generated for an EOC molecular landscape. Data analyses were relatively superficial, and some analyses were also flawed and not correct. Overall, the manuscript did not provide biological insight into the EOC, and the major claimed discoveries including molecular signatures and therapeutics targets were not truly validated.

(1) CPTAC already generated two comprehensive proteomics data. The discovery cohort (Zhang et al., 2016, Cell) included 160 high-grade serous carcinomas (HGSC) and the validation cohort (McDermott et al, 2020, Cell Reports Medicine) generated proteomics data from 83 prospectively collected ovarian HGSC. The question for the current manuscript is what new findings were discovered from the cohort with proteome data only compared with comprehensive proteogenomic data from two CPTAC cohorts. Can the major findings be validated in CPTAC cohorts? If not, what's the explanation? The authors should address the main motivations of this study. Why is it different from CPTAC studies? Actually, the CPTAC discovery study (Zhang et al., 2016, Cell) was not even mentioned in this manuscript. There was no clinically meaningful focus in the manuscript. For example, after CPTAC studies, the research communities are very interested in the new biomarkers and therapeutic targets for chemotherapy resistance of EOC. However, the current study is still focusing on differently expressed protein list generation.

(2) A lot of crucial genomic information was missing for this "proteogenomic" study. The major finding and conclusions of this manuscript were basically a differential protein expression list. For a more comprehensive study, the following genomic data need to be generated:

(a) mutation and CNV data. It may be difficult to generate whole exome sequencing data, but targeted sequencing for EOC common mutations/CNVs is necessary.

(b) transcriptome data. Technically it's not difficult to generate the whole transcriptome data from FFPE samples, and it's very important to integrate the transcriptome data and proteome data to understand the molecular changes of EOC, and to reduce possible false discoveries.

(c) Ideally, both phosphoproteome and acetylome data, or at the least phosphoproteome data need to be generated. The RNA, protein, and phosphorite levels data are needed to describe changes in important oncogenic signaling pathways, in order to understand the biology underlying recurrence and treatment resistance. Overall very limited data (and analysis) was added by this study as a proteogenomic characterization paper, and it's quite difficult to generate biological hypotheses from proteome data only.

(3) There was no multiple comparison adjustment for several important analyses. For example, the functional enrichment analysis on Page 7, the identification of differentially expressed proteins among three histological subtypes on Page 6, and the survival analysis report in SupTable 3. For high-dimensional genomic/proteomic data, it's easy to have false positive findings. The adjusted p-values should be used instead of nominal p-values.

(4) The Kolmogorov-Smirnov test on Page 6 may not be correct since the KS test is not for testing multiple groups.

(5) For the description and analysis of clinical data from three subtypes in 3.1, the results may be biased due to data collection. Are the results consistency with the previous studies? There were no literature reviews and discussions. The clinical data including age, stage, survival outcomes, etc of each individual patient should be included in the supplementary data.

(6) The complete protein and pathway analysis ranking list including differentially expressed outcomes, functional enrichment analysis, and survival analysis should be listed in the supplementary data.

(7) In the result section 3.2, the discovery and validation of diagnostic makers may not be meaningful since it's unlikely to use protein biomarkers for diagnostic purposes.

(8) There was no validation for the biomarker discovery from section 3.4

(9) There was no solid foundation to claim the potential therapeutic targets in section 3.6. Those proteins are just top tanking candidates from differentially expressed protein lists and/or survival analysis lists. Nothing was related to the true biological mechanisms. The mechanism experiments including cell lines and animal experiments need to perform to confirm the findings.

Reply to reviewer #1 (expertise in proteogenomics and cancer):

The paper entitled Proteogenomics Characterization of Epithelial Ovarian Cancer Delineates Molecular Signatures and Therapeutic Targets in Distinct Pathological Subtypes by Gong et al. aims to profile the proteomic features of Epithelial Ovarian Cancer with different histological stages and to integrate this information with patients' prognosis. The focus of the authors was on identifying proteomic signatures that associates with patients' survival. Overall, the paper provides the resource to the proteomics and EOC communities and fits the scope of Nature Communications, however there are some concerns need to be satisfactorily addressed.

Major points:

Q1. The title of the paper needs to be changed, since the author did not perform genomic analysis.

REPLY: We thank the reviewer for this careful consideration. We accepted your suggestion and amended the title to “Proteomic Characterization of Epithelial Ovarian Cancer Delineates Molecular Signatures and Therapeutic Targets in Distinct Histological Subtypes”.

Q2. It was not fully clear how these samples are actually collected, evaluated and processed to ensure are certain degree of tumor cellularity?

REPLY: We thank the reviewer's suggestion. We have added detailed information on the sample collection, assessment, and processing in the main text and supplementary materials, with the following details:

On page4:

A total of 239 formalin-fixed, paraffin-embedded (FFPE) ovarian epithelial tissues were acquired from newly diagnosed EOC patients undergoing primary debulking surgery at the Shengjing Hospital of China Medical University (Shenyang, China) from 2013 to 2019. **Detailed information on sample collection, evaluation, and processing is in the supplementary materials.** All patients were treated with both carboplatin and paclitaxel. The 239 patients were included in the present analysis with three histological types: SC samples (n = 80), EC samples (n = 79) and CCC samples (n = 80). Based on the characteristics of EOC, it is extremely difficult to obtain para-carcinoma tissue ¹. **Therefore, the histologically normal ovarian tissues (n = 30) taken from cases of uterine fibroids were used as CT samples, in which the ovary was surgically removed incidental to radical surgery. The median age of patients in the CT samples at time of surgery was**

55 years old and ranged between 41 and 71 years old. All of these samples were examined by two experienced pathologists who confirmed the diagnosis of the disease samples. Eligible tumor samples included at least 90% tumor cells.

On Supplementary Materials:

1. Formalin Fixed, Paraffin Embedded (FFPE) ovarian tissue section preparation

(1) Obtaining a fresh specimen

Cut small blocks of tissue $1 \text{ cm}^2 \times 0.4 \text{ cm}$, and place them in a histological/tissue processing cassette.

Cautious:

- The ovarian tissue was removed gently to avoid trauma by an expert gynecologist.
- Specimen is not allowed to dry out prior to fixation.
- Avoid contaminating fresh specimens with foreign chemicals or substances such as disinfectants.
- Each specimen should be properly identified and name, pathology number, and other details recorded as soon as possible.
- Fixation is always carried out promptly. If it is necessary that a specimen remains unfixed for a short period of time, it should be refrigerated at 4 °C.

(2) Fixation

To the tissue, add 20× the tissue volume 10% neutral formalin.

Cautious:

- The specimen is placed in formalin, this will slowly penetrate the tissue causing chemical and physical changes that will harden and preserve the tissue and protect it against subsequent processing steps.
- An adequate volume of fixative (ratio of at least 20:1) is used in a container of an appropriate size. This avoids distortion of the fresh specimen and ensures good quality fixation.
- Ideally, specimens should remain in fixative for long enough for the fixative to penetrate every part of the tissue and then for an additional period to allow the chemical reactions of fixation to reach equilibrium (fixation time). Generally, this will mean that the specimen should fix for between 6 and 48 hours.

(3) Dehydration

Because melted paraffin wax is hydrophobic (immiscible with water), most of the water in a specimen must be removed before it can be infiltrated with wax, a typical dehydration sequence for specimens not more than 4mm thick would be:

- (1) 80% ethanol 1h
- (2) 90% ethanol 1h
- (3) 95% ethanol 1h
- (4) 95% ethanol 1h
- (5) 100% ethanol 1h
- (6) 100% ethanol 1h
- (7) 100% ethanol 1h

Cautious:

- Processing reagents are replaced strictly according to established guidelines.

(4) Clearing

A popular clearing agent is xylene, and multiple changes are required to completely displace ethanol, a typical clearing sequence for specimens not more than 4mm thick would be:

- (1) xylene 1h
- (2) xylene 1h

Cautious:

Processing reagents are replaced strictly according to established guidelines.

(5) Wax infiltration

A typical infiltration sequence for specimens not more than 4mm thick would be:

- (1) wax 1h
- (2) wax 1h
- (3) wax 1h

Cautious:

- High quality wax is used for infiltration to ensure high quality blocks that are easy to cut.

(6) Embedding

This step is carried out using an “embedding centre” where a mold is filled with molten wax and the specimen placed into it. The specimen is very carefully orientated in the mold because its placement will determine the “plane of section”, an important consideration in both diagnostic and research histology. A cassette is placed on top of the mold, topped up with more wax, and the whole thing is placed on a cold plate to solidify. When this is completed, the block with its attached cassette can be removed from the mold and is ready for microtomy. It should be noted that, if tissue processing is properly carried out, the wax blocks containing the tissue specimens are very stable

and represent an important source of archival material.

Cautious:

- Specimens are carefully orientated, competent grossing ensures flat surfaces on most specimens.
- A mold of suitable size is always chosen for each specimen.
- Specimens are handled gently during embedding.
- Before handling tissue, forceps are heated to the point where the wax just melts.
- Before handling tissue, forceps are heated to the point where the wax just melts.
- Molds are filled to an optimum level and do not overflow.

Q3. Also how did the NAT samples were chosen, what criteria did the author apply?

REPLY: We thank the reviewer's suggestion. The normal ovarian tissues (n = 30) were taken from cases of uterine fibroids, in which the ovarian was surgically removed incidental to radical surgery for non-ovarian diseases. The median age of women at time of surgery was 55 years old and ranged between 41 and 71 years old.

On Page4:

Based on the characteristics of EOC, it is extremely difficult to obtain para-carcinoma tissue¹. Therefore, the histologically normal ovarian tissues (n = 30) taken from cases of uterine fibroids were used as CT samples, in which the ovary was surgically removed incidental to radical surgery. The median age of CT samples at time of surgery was 55 years old and ranged between 41 and 71 years old.

Q4. The paper lack of details on the quantification of proteomic data. It is unclear how many proteins were detected in each sample. Also, did the author utilize imputation strategy for missing values?

REPLY: We thank the reviewer for this careful consideration. As per your suggestion, we have added the details on the quantification of proteomic data and count the number of detected proteins Supplementary Figure 1. The number of proteins we detected is consistent with previous studies^{2,3}, and also showed a significantly higher number of proteins identified in the tumors than in the control tissues⁴. In addition, proteomic data were filtered to remove proteins with missing values from more than half of the samples⁴. We also used an imputation strategy for missing values, using the DreamAI algorithm to estimate missing values⁵.

On Page6:

2.3 Proteomic Data Filtering and Normalization

Label-free quantitation (LFQ) intensity of 269 samples (239 EOC and 30 CT samples) were obtained from the Maxquant result files. Proteins with missing values in more than half of the samples were removed⁴. As a result, 4,447 proteins out of a total of 8,257 proteins were retained. The LFQ intensity of the 4,447 proteins was normalized using the normalized quantile functions in the R package ‘limma’⁶. Missing values were imputed using the DreamAI algorithm⁵.

On Page9:

In the current cohort, proteome analysis was performed using label-free technology on the same mass spectrometer with consistent quality control (Figure 1A, see the ‘Methods’ section). A total of 8,257 proteins were identified across all tumor samples. The number of proteins detected in each sample was shown in Supplementary Figure 1A, in which Supplementary Figure 1B demonstrated the details of the proteomic data quantification for each group.

Supplementary Figure 1. Detected proteins in each sample. A. Bar plot showing the number of detected proteins in each sample, colored by the EOC subtypes. **B.** Box plot showing the number of detected proteins across EOC subtypes.

Q5. Moreover, the author integrated phosphoproteomic data from previously published paper by CPTAC, while direct comparison/integration to the CPTAC cohort may be challenging due to the different proteomics technologies used in both studies. Are the baseline characteristics of patients in the current cohort different from CPTAC cohort? Did the difference impact patients’ prognosis and further analysis?

REPLY: We thank the reviewer for this careful consideration. The CPTAC stores raw mass spectrometry-based data files for a variety of cancers (e.g., ovarian cancer) to accelerate understanding of the molecular basis of cancer through the application of

large-scale proteomic and genomic analysis. In the ovarian cancer data from CPTAC, the patients were predominantly white and had advanced tumor staging (Stage III and IV). And, the age of patients was 59.4 ± 10.7 years, where the maximum value was 86 and the minimum value was 37⁷. In our study, the patients were Asian population, and had an age of 53.26 ± 9.70 years, where the maximum value was 80 and the minimum value was 26. As the patients with epithelial ovarian cancer in the two datasets belonged to different races/ethnicities, the patients showed some differences at the baseline characteristics. Previous studies have also demonstrated that Asian patients were younger than white patients in the Surveillance, Epidemiology, and End Results (SEER) program⁸, reflecting the difference in mortality among patients with epithelial ovarian cancer by race/ethnicity. This is a limitation of our analysis by integrating phosphoproteomic data from previously published paper by CPTAC, and an exploration of its limitations has been added to the Discussion section (**See: Page 18**).

However, at the molecular level, there is some similarity between our dataset and the CPTAC cohort⁷. For example, multiple proteins (e.g., COL1A1, COL1A2, ITGA2B, ITGB3, etc.) are significantly down-regulated in both datasets, as well as multiple identical biological processes (e.g., humoral immune response, focal adhesion, regulation of endocytosis, etc.) in which dysregulated proteins are significantly enriched. In addition, multiple prognostic markers identified in our study could be validated in the CPTAC cohort, such as COL1A2 and IFIT3. The consistency of these proteomic characteristics also supports the integration of phosphorylation data from the CPTAC cohort for multi-perspective analysis.

On page18:

We recognize several important limitations. First, historical epidemiologic data has suggested that the incidence and survival rates of OC depends on the ethnicity and geographical area^{9,10}. Since East Asian backgrounds were significantly younger compared to other races and have an earlier stage of OC^{11,12}. This could contribute to the fact that our cohort had favorable clinical outcomes than that of the general patients¹³. However, since our study only included patients from China, the data may not fully represent the entire population. This inevitably limits generalization to other populations and introduces the possibility of bias towards particular demographics¹⁴. Then, we integrated phosphoproteomic data from a previously published paper by CPTAC to investigate the important role of proteins and their post-translational modifications in signaling pathways. Due to differences in patient race/ethnicity (the

patients in our study were yellow, whereas the patients in CPTAC were predominantly white), this may have imposed some limitations on the use of phosphoproteomic data. Further research is necessary to investigate and confirm the findings reported hereof to be clinically meaningful.

Q6. The comparison to other published data sets is limited. This is important because this study is based on FFPE archival samples and not fresh frozen tissues. Please comment? Also, how comprehensive and correct is the presentation of proteomics based on FFPE. It would be important to validate some of the findings in this study rather than adding new omics data.

REPLY: We thank the reviewer's suggestion. Normal frozen tissue is not suitable for storage for long periods of time, but FFPE samples can be stored for decades. Formalin can induce the cross-linking of proteins and thereby prevent protein degradation, which is the main advantage of FFPE sample¹⁵. Many clinical samples take a very long time to collect and may take years to collect, therefore paraffin embedding is the most used method of clinical sample preservation and is more suitable for the long-term preservation of clinical samples¹⁵. The feasibility of paraffin samples for proteomic has been reported in a number of articles, such as the study "Proteomics of Melanoma Response to Immunotherapy Reveals Mitochondrial Dependence" by Michal Harel et al. published in Cell in 2019², and the study "Multi-organ proteomic landscape of COVID-19 autopsies" by Xiu Nie et al. published in Cell in 2021³. These illustrate the availability of proteomic based on FFPE samples.

In addition, we review several previous studies that have portrayed the molecular composition and underlying mechanisms of ovarian cancer based on proteomic signatures. For example, Zhang et al. delineated the pathways and processes that drive the biology of high-grade serous ovarian cancer (HGSOC) and how these pathways are altered relative to the clinical phenotypes⁷. And, McDermott et al. demonstrated that proliferation-induced replication pressure promotes characteristic chromosomal instability in HGSOC¹⁶. The results of our study are consistent with these previous studies at various dimensions. Firstly, at the protein abundance level, we found that most of the dysregulated proteins were down-regulated in expression in tumor tissues. For example, proteins involved in ECM-receptor interactions (COL1A1, COL1A2, COL2A1, COL4A1, COL4A2, COL6A1, COL6A2, COL6A3, LAMA2, LAMA4, LAMB2, etc.), as well as proteins involved in the PI3K-AKT signaling pathway

(ITGA2B, ITGB3, etc.) tended to be down-regulated both in our study and in the study by McDermott et al. ¹⁶. Then, at the level of biological function, the down-regulated proteins were all significantly enriched in biological processes, such as humoral immune response, focal adhesion, and regulation of endocytosis ¹⁶. Additionally, at the level of prognostic value, our study found that patients with high COL1A2 and IFIT3 expression had a worse prognosis (log-rank P-values < 0.05), which was also validated in the CPTAC cohort. These consistent results significantly increase the credibility of our study, and reaffirm the usability of FFPE sample-based proteomic.

Finally, a comparison of our findings with previous studies has been added to the Introduction and Discussion section of the manuscript.

On Page3:

Furthermore, several studies have explored the important role of proteomic profiling in high-grade serous ovarian cancer (HGSOC). The study by Zhang et al. provided additional insights into the pathways and processes that drive the biology of HGSOC and how these pathways are altered relative to the clinical phenotypes ⁷. And, McDermott et al. described a potential role for proliferation-induced replication stress in promoting the characteristic chromosomal instability of HGSOC ¹⁶. A study from Coscia et al. also revealed that cancer/testis antigen 45 (CT45), a prognostic factor associated with the doubling of disease-free survival, enhanced chemosensitivity in metastatic HGSOC ¹⁷. However, molecular characterization of EOC histological subtypes (CCC, EC, and SC) using large cohorts is still limited.

On Page17:

Notably, most dysregulated proteins tended to be under-expressed in tumor tissues, which was consistent with a previous study ¹⁶. For example, the expression abundance of multiple key proteins (such as COL1A1, COL1A2, ITGA2B, ITGB3, etc.) in the ECM-receptor interaction or PI3K-AKT signaling pathways was significantly down-regulated ¹⁶. And, significant enrichment of biological functions, such as humoral immune response, focal adhesion, and regulation of endocytosis, was corroborated ¹⁶.

Q7. Functional experiments should be done, since the current study is based on bioinformatic analysis, it is important for the author to perform functional analysis utilizing PDCs and PDX models.

REPLY: We thank the reviewer for this careful consideration. Recent improvements in mass spectrometry-based proteomic now enable direct examination of the

consequences of genomic aberrations, providing deep and quantitative characterization of tumor tissues¹⁸. Proteomic is already leading to new biological and diagnostic knowledge with the potential to improve our understanding of malignant transformation and therapeutic outcomes. Moreover, our findings can be validated in the CPTAC cohort, e.g., dysregulated proteins (COL1A1, COL1A2, ITGA2B, and ITGB3), prognostic markers (e.g., COL1A2 and IFIT3), etc. These results confirm to some extent the stability of our findings.

In addition, we found the specific prognostic value of MPP7 in serous carcinoma in Result 3.6. Prognostic analysis of the CPTAC cohort also demonstrated that high MPP7 expression was associated with poor overall survival in serous carcinoma patients. Based on the prognostic value of MPP7 and the degree of malignancy of serous carcinoma, we performed in vitro experiments to validate the biological function of MPP7 in serous carcinoma cells. CCK-8 assays showed that shRNA-mediated MPP7 knockdown decreased the proliferation of serous carcinoma cells. Flow cytometric analysis of cell cycle demonstrated that MPP7 knockdown resulted in decreases in G1-S transition in serous carcinoma cells. Flow cytometric analysis of cell apoptosis elucidated that serous carcinoma cell apoptosis was induced by MPP7 knockdown. Transwell assays showed that serous carcinoma cell migration and invasion were inhibited by MPP7 knockdown. These results confirmed the important role of MPP7 in SC progression and metastasis.

On Page15:

Based on the prognostic value of MPP7 and the degree of malignancy of SC, we investigated the function of MPP7 in SC cells in vitro. Malignant behaviors including cell proliferation, cell migration, and cell invasion, cell cycle distribution, and cell apoptosis were assessed. CCK-8 assays showed that shRNA-mediated MPP7 knockdown decreased cell viability (Supplementary Figure 7A and Supplementary Table 8), indicating the inhibition of cell proliferation by MPP7 knockdown. Flow cytometric analysis of cell cycle demonstrated that MPP7 knockdown resulted in decreases in cells at the S phase and increases in cells at the G1 and G2 phase (Figure 6G and Supplementary Table 9). The results implied that MPP7 was involved in regulation of G1-S and S-G2 transition. Flow cytometric analysis of cell apoptosis elucidated that cell apoptosis was induced by MPP7 knockdown (Figure 6H and Supplementary Table 10). Transwell assays suggested that cell migration and invasion were inhibited by MPP7 knockdown (Supplementary Figure 7B&C and Supplementary

Table 11&12). These findings uncovered that the biological function of MPP7 in SC. Taken together, we propose further analysis of these subtype-specific proteins as promising therapeutic targets in the three histological subtypes.

Figure 6. Potential therapeutic targets for distinct histological subtypes.

A, C, E. Expression levels of CSPG4, TMEM87A, and MPP7 among the respective histological subtypes. **B, D, F.** Kaplan-Meier survival curves for patients expressing CSPG4, TMEM87A, and MPP7 in CCC, EC, and SC patients, respectively. Among them, the red line represents highly expressed protein and the blue line represents lowly

expressed protein. **G.** OVCAR-3, A2780, and ES-2 cells were infected with lentiviral vectors carrying shMPP7 or shNC. Cell cycle distribution was analyzed by flow cytometry at 48 h after infection. The gating strategy was shown in Supplementary Figure 8A. **H.** Cell apoptosis was measured by Annexin V-FITC/PI staining at 48 h after infection. The count of Annexin V-FITC-positive and PI-positive cells (Late apoptosis) and Annexin V-FITC-positive and PI-negative cells (Early apoptosis) was assessed by flow cytometry. The gating strategy was shown in Supplementary Figure 8B. **P-value < 0.01 and **** P-value < 0.0001 vs. the shNC group (n=4); data were analyzed by the one-way analysis of variance (ANOVA) followed by Tukey's tests or the Brown-Forsythe and Welch ANOVA tests followed by Tamhane T2 tests.

Supplementary Figure 7. Effect of MPP7 knockdown on the malignant behavior of epithelial ovarian cancer cells. OVCAR-3, A2780, and ES-2 cells were infected with lentiviral vectors carrying shMPP7 or shNC. **A.** CCK-8 assays were performed at 0, 24, 48, 72, and 96 h after infection. **B-C.** Cell migration (B) and invasion (C) were determined by transwell assays at 48 h after infection. Scale bar: 100 μ m. * P-value < 0.05, **P-value < 0.01, and **** P-value < 0.0001 vs. the shNC group (n=4); data were analyzed by the one-way or two-way analysis of variance (ANOVA) followed by Tukey's tests.

Supplementary Figure 8. Gating strategies for flow cytometric analysis. A. Gating strategies for cell cycle analysis. The major cell population was selected based on FSC and SSC, and then the single cells were gated by PI-A and PI-H. The signal of PI in the single-cell population was analyzed. **B.** Gating strategies for cell apoptosis analysis (Annexin V-FITC-PI staining). Cells were examined by FSC and SSC to obtain the major cell population. Then, the signal of FITC and PI in the cell population was analyzed.

On Page8:

2.9 Experimental methods and statistical analysis

Cell culture and infection Ovarian cancer cell lines OVCAR-3, A2780, and ES-2 were cultured in RPMI-1640 medium (Solarbio, China) containing 20% FBS (TIANHANG, China), RPMI-1640 medium containing 10% FBS, and McCoy's 5A medium (Procell, China) containing 10% FBS at 37°C with 5% CO₂, respectively. Cells were infected with lentiviral vectors carrying short hairpin RNA (shRNA) targeting MPP7 (shMPP7) or its negative control shRNA (shNC) (OVCAR-3: MOI = 30; A2780: MOI = 10; ES-2: MOI = 5).

Cell counting kit-8 (CCK-8) assay Cells (3×10^3) were seeded onto a 96-well plate and harvested at 0, 24, 48, 72, or 96 h after infection. The CCK-8 assays were performed using the CCK-8 assay kit (Wanleibio, China) according to the manufacturer's instructions. The optical density (OD) was detected at 450 nm using a BioTek 800 TS plate reader (Agilent, USA).

Flow cytometry Cell cycle distribution and cell apoptosis were analyzed using flow cytometry. For cell cycle analysis, cells were fixed with 70% cold ethanol and washed with PBS at 48 h after infection. After incubation with RNase A at 37°C for 30 min, the

cells were stained with Propidium Iodide (PI) for 30 min. For cell apoptosis detection, cells were washed with PBS at 48 h after infection and resuspended in binding buffer. Then, the cells were stained with Annexin V-FITC and PI for 15 min. The Cell Cycle Analysis Kit and the Annexin-FITC-PI Staining Kit were purchased from Wanleibio (China). The signal of PI and Annexin V-FITC was detected by a NovoCyte flow cytometer (Agilent, USA). The results obtained from flow cytometric analysis were analyzed by the NovoExpress software (version 1.4.1, Agilent, USA).

Transwell assay The transwell chambers used for migration assays or the matrigel-coated transwell chambers used for invasion assays were placed into 24-well plates. Cells were resuspended in serum-free medium at 48 h after infection and seeded in the transwell upper chamber at 5,000 (Migration assay) or 50,000 (Invasion assay) cells per well. After 24 h, the cells on the lower side of the transwell membrane were fixed with 4% PFA (Aladdin) for 20 min, then stained with 0.5% crystal violet (Amresco, USA) for 5 min, and counted under a microscope (OLYMPUS, Japan).

Statistical analysis GraphPad Prism 8 (GraphPad Software, USA) was used for statistical analyses. The Shapiro-Wilk normality test and the Brown-Forsythe test were used for analysis of normal distribution and variance homogeneity, respectively. Data, which were normally distributed and had equal variances, were analyzed using one-way or two-way analysis of variance (ANOVA) followed by Tukey's tests. Data, which were normally distributed and had unequal variances, were analyzed using Brown-Forsythe and Welch ANOVA tests followed by Tamhane T2 tests. Data are presented as mean \pm standard deviation (SD). A P-value < 0.05 was considered statistically significant.

Q8. Also, the research lacks of validation either at functional experiment level or at cohort level. It is interesting that the author conducted PRM analysis to verify proteomic biomarkers, however, the accuracy and applicable of those proteins need to be further confirmed in an independent validation cohort.

REPLY: We thank the reviewer for this careful consideration. We validated the results of this study in the CPTAC cohort ¹⁶, which were consistent across multiple outcomes. Firstly, most of the proteins with dysregulated expression in both datasets were down-regulated in tumor tissues, such as multiple collagen family proteins. Then, the down-regulated proteins were all significantly involved in humoral immune response, focal adhesion and regulation of endocytosis, and many other important biological processes.

Finally, the prognostic power of multiple prognostic markers (e.g., COL1A2, IFIT1, IFIT2, and IFIT3) was also validated in the CPTAC cohort.

On Page17:

Notably, most dysregulated proteins tended to be under-expressed in tumor tissues, which was consistent with a previous study ¹⁶. For example, the expression abundance of multiple key proteins (such as COL1A1, COL1A2, ITGA2B, ITGB3, etc.) in the ECM-receptor interaction or PI3K-AKT signaling pathways was significantly down-regulated ¹⁶. And, significant enrichment of biological functions, such as humoral immune response, focal adhesion, and regulation of endocytosis, was corroborated ¹⁶.

Q9. Both raw data and processed data from transcriptomic and proteomic analysis should be made available in public repositories such as Scientific Data.

REPLY: We thank the reviewer's suggestion. Our data were uploaded to the public repository. The mass spectrometry proteomic data have been deposited to the ProteomeXchange Consortium (<http://www.proteomexchange.org/>) via the PRIDE partner repository with the identifier PXD033741. In addition, we obtained the necessary approval from the Chinese Ministry of Science and Technology (MOST, <https://grants.most.gov.cn/grants/>) related to export the genetic information and materials related to this work (registration number: 2023BAT0). The mass spectrometry proteomic data also have been deposited to the Open Archive for Miscellaneous Data (OMIX, <https://ngdc.cncb.ac.cn/omix/>) with the identifier OMIX002719.

On Page 20:

Data and materials availability

The mass spectrometry proteomic data have been deposited to the ProteomeXchange Consortium (<http://www.proteomexchange.org/>) via the PRIDE partner repository with the identifier PXD033741. The mass spectrometry proteomic data also have been deposited to the Open Archive for Miscellaneous Data (OMIX, <https://ngdc.cncb.ac.cn/omix/>) with the identifier OMIX002719.

Q10. All statistical analysis should be checked by statisticians.

REPLY: We thank the reviewer's suggestion. We carefully checked and revised the statistical analysis involved in our study. For example, differential expression analysis (whether there were differences in protein abundance between cancer and control groups or between the three histological subtypes), functional enrichment analysis, and

survival analysis in the study were corrected by multiple hypothesis testing. In addition, a descriptive mistake was found in the identification of specific proteins for the three histological subtypes. The "Kolmogorov-Smirnov test" should be amended to "Kruskal-Wallis test". The Kruskal-Wallis test was used to determine whether there was a statistically significant difference between the medians of the three or more independent groups.

On Page6:

2.4 Differential Expression Analysis of the Proteome

Differential proteomic analysis was conducted on 4,447 quantifiable proteins in a total of 8,257 proteins detected. We performed Wilcoxon signed-rank tests to identify the dysregulated proteins with a statistically significant P-value between EOC and CT patients. The P-values were corrected by the **Benjamini-Hochberg (BH) procedure**. Significantly up- or down-regulated proteins were extracted by a threshold of **adj.P-value** < 0.01 and $|\log_2(\text{fold change})| > 1$.

To assess differentially expressed proteins across treatment groups, **Kruskal-Wallis test (K-W test)** was used to identify differentially expressed proteins among the three histological subtypes of EOC (SC, EC, and CCC). Post-hoc tests were performed to identify the differentially expressed proteins between any two subtypes (**adj.P-values** < 0.05, R package 'PMCMRplus').

2.5 Survival Analysis

For the clinicopathological analysis, Fisher's exact test (two-sided) was performed. Kaplan–Meier curves and log-rank tests were used to compare OS or RFS among the proteomic subtypes. Clinical associations of protein expression was examined by the Cox proportional hazards model, **and P-values were corrected by the False Discovery Rate (FDR) procedure**. Univariable and multivariable Cox regressions were applied to estimate the hazard ratios (HR), 95% confidence intervals (CI), **Cox P-values and Cox adj.P-values** of each protein.

2.6 Functional Enrichment Analysis

Comprehensive function annotation of proteins, including Gene Ontology (GO) ^{19,20}, Kyoto Encyclopedia of Genes and Genomes (KEGG) pathways ²¹ and Gene Set Enrichment Analysis (GSEA) ²², was performed on R package 'clusterProfiler' ²³, Metascape (<http://www.metascape.org/>) and R package 'fgsea' ²⁴ to identify GO biological processes (BPs), KEGG pathways, Reactome gene sets ²⁵, WikiPathways ²⁶, and Hallmark gene sets ^{22,27} in which dysregulated proteins were enriched. The **adj.P-**

values < 0.05 were considered statistically significant. R packages ‘simplifyEnrichment’²⁸ and ‘GOSemSim’²⁹ were used to cluster GO terms based on similarity matrices of functional terms. The function ‘simplify’ of R package ‘clusterProfiler’ was used to remove redundancies of enriched GO results.

Reply to Reviewer #2 (expertise in ovarian cancer TME and metabolism):

This paper comprehensively analyzes novel insights into the 580 biological characterization for the improvement of clinical diagnosis and therapeutics of EOC. It is my opinion that this will contribute significantly to this field due to its novel and thorough approach.

Authors provide sufficient evidence to support their claims.

Methodology is sound and meets the criteria of the standards in our field. Figures and legends are well done and discussion underscores importance of their findings.

Q1. Minor critique: the figures 4 and 5 have very small sized font, please consider increasing font or rearranging figure so as not to be so filled with words (can be overwhelming and hard to follow).

REPLY: We sincerely appreciate this positive feedback and have revised the font size and alignment of Figure 4 and Figure 5.

Figure 4. Protein co-expression network and tumor progression landmarks. A. Protein co-expression network of 896 nodes and 13,574 edges. Nodes are color coded according to module membership. Representative enriched biological terms are shown for distinct modules. **B.** Density plots of the pairwise protein-protein correlations for the interactions shown in the network. **C.** Aberrant protein expression levels were superimposed on the network for each histological subtype. The red and blue dots represent up- and down-regulated differentially expressed proteins, respectively. **D.** Sub-network of Module31. **E.** Boxplots illustrated the abundances of IFIT1, IFIT2 and IFIT3 in the different histological subtypes. **F.** Kaplan-Meier plots for IFIT3.

Figure 5. Comparison and characterization of subtype-specific proteins. **A.** Venn diagram shows the overlap of differentially expressed proteins reported in different subtypes of EOC. **B-C.** GO categories and molecular pathways enriched in differentially expressed subtype-specific proteins. GO categories were grouped according to functional theme. **D.** Subtype-specific proteins were enriched in the pathway of Signaling by Rho GTPases.

Reply to Reviewer #3 (expertise in epithelial ovarian cancer):

Proteogenomics Characterization of Epithelial Ovarian Cancer Delineates Molecular Signatures and Therapeutic Targets in Distinct Pathological Subtypes by Gong et al aims to characterize the proteomic landscape of different subtypes of ovarian cancer and the diagnostic and prognostic value of these proteins.

The authors used 239 cancer samples and 30 normal control ovarian tissue to characterize the different proteomes and to discover aberrant pathways. The main strength of this paper is the large sample size, the use of negative control ovarian samples and the comprehensive proteomics analysis. In addition, the paper identifies several pathways which could be potentially used as future targets in ovarian cancer.

Questions to the authors:

Q1. *In the methods section, the authors state that a total of 239 formalin-fixed, paraffin-embedded (FFPE) ovarian epithelial tissues were acquired from newly diagnosed EOC patients undergoing primary debulking surgery at the Shengjing Hospital of China Medical University (Shenyang, China) from 2013 to 2019. The last survival follow-up data on these patients were in March 2021. However, based on the Supplementary figure – half of the serous cancer patients were still alive, and only about 40% of the endometrioid and clear cell patients had recurred. This seems to be a much more favorable outcome than one would be expected in a general OC population. How do the authors explain this difference?*

REPLY: We thank the reviewer for this careful consideration. Stark racial/ethnic disparities in ovarian cancer (OC) diagnosis age and survival outcomes have been well documented by previous analysis^{8,12,13,30-33}. Consistent with these earlier studies, we found that women in our cohort tended to be younger, diagnosed at an earlier stage, and had more favorable outcome, compared to general OC population. Our results agree

with the several reports evaluating OC survival disparities in Asians. For example, Fuh et al. published results in OC patients between white and Asian women from the Surveillance, Epidemiology, and End Results (SEER) program over a 21-year period ⁸. Similarly, they found that Asian patients were younger than White patients. Additionally, they determined that Asian women maintained a survival advantage after adjusting for age, stage, histology, and extent of surgical treatment. The authors also divided Asian women into ethnic subgroups and observed that Chinese, Korean, Filipino, and Vietnamese women had better survival in comparison to Japanese and Indian/Pakistani women, which remained statistically significant in multivariate analysis ¹². Multiple factors contribute to OC-related mortality and to racial/ethnic disparities. Allelic variants in genes encoding important drug metabolizing enzymes have been detected in Asians such as the CYP3A4 and CYP3A5 gene that is involved in the metabolism of numerous anticancer drugs including docetaxel and paclitaxel ³⁴. Younger age at diagnosis, and thus, better physical fitness and higher tolerance to aggressive treatments, may be another possible explanation for this disparity ^{30,31}. These may partly explain the better survival rate of the patients in our study compared to a general epithelial ovarian cancer population.

On page16:

In clinical characteristics, we found that women in our cohort tended to be younger and had a more favorable outcome compared to the general EOC population, which is consistent with several previous studies ^{8,12,13,30-33}. In terms of age, previous evidence has shown that pregnancy history has been consistently related to OC risk ³⁵. As the fertility rates in China have continued to change in recent years ³⁶, this may partly contribute to the early development of OC. Younger age at diagnosis, and thus, better physical fitness and higher tolerance to aggressive treatments, may be another possible explanation for this disparity ^{30,31}.

Q2. As much of the proteomics analysis is based on RFS/OS analysis, less than half of the cohort progressed, and 75% of the cohort is still alive. Do the authors have more updated survival data on this cohort of patients?

REPLY: We thank the reviewer's suggestion. With survival follow-up data as of March 31, 2021, the study cohort had a mortality rate of 33.89% and a recurrence rate of 57.32%. Based on the comments of the reviewers, we updated the patient survival data

with a final follow-up of December 31, 2021. With survival follow-up data as of December 31, 2021, the study cohort had a mortality rate of 43.93% and a recurrence rate of 63.17%. Comparing the two follow-up time points, the number of deaths and recurrences was slightly increased, but the mortality rate still did not reach half in a general epithelial ovarian cancer population, which may be related to the diagnosis age and race of the patients in our study. In addition, we set the follow-up time to December 31, 2021, and updated the outcomes related to survival and recurrence in this study. The follow-up time either as of March 2021 or December 2021, the important biomarkers identified in our study showed significant association with overall survival and recurrence survival, such as CDK4, CDKN1B, COL1A2, etc., which indicated the stability of these biomarkers.

Figure 3. Signaling pathway disturbances in EOC suggest therapeutic opportunities. **A.** Protein interactions from the KEGG pathway. **B.** Heatmap showing the expression of proteins in four pathways. **C.** Forest plots show HRs of proteins using Cox analysis. Among them, asterisks represent Cox adj.P-values, *adj.P-values < 0.05.

Q3. The authors comment on the heterogeneity of OC, although in their study, they mainly compare different subtypes to each other. Therefore, it is not surprising that the different subtypes have a distinct proteomic landscape. Did the authors look at the difference in the same subtype in early-stage disease (stages 1 and 2) vs. late-stage (stages 3&4)?

REPLY: We thank the reviewer for this careful consideration. Based on the reviewers' comments, we investigated the differences in protein and molecular functions in early-stage (stage I and II) and late-stage (stage III and IV) of tumors in three histological subtypes, respectively. Firstly, we compared the abnormal abundance of proteins during tumor progression (late-stage group vs. early-stage group) in three histological subtypes, respectively. We identified differentially expressed proteins during tumor progression stages in each subtype, as well as revealed these aberrant proteins with subtype specificity (Supplementary Figure 6A). Then, the molecular functions involved in tumor progression biomarkers in the three histological subtypes were also identified separately (Supplementary Table 7). The results suggested that each histological subtype has specific KEGG signaling pathways, which promote tumor progression (Supplementary Figure 6B). In particular, the serous carcinoma subtype was involved in abnormalities of multiple signaling pathways, mainly for cellular processes (focal adhesion, regulation of actin cytoskeleton, phagosome, tight junction, endocytosis, and adherens junction) and organismal systems (leukocyte transendothelial migration, platelet activation, neurotrophin signaling pathway, Fc γ R-mediated phagocytosis, and chemokine signaling pathway) (Supplementary Table 7). Among them, focal adhesion promotes tumor progression and metastasis through effects on cancer cells and stromal cells of the tumor microenvironment³⁷, and leukocyte transendothelial migration is essential for immune surveillance and inflammatory responses^{38,39}. In addition, maintenance of genomic integrity is one of the fundamental features of life⁴⁰, where DNA replication is tightly regulated to maintain genomic stability⁴¹. When these regulatory mechanisms fail, replication stress and DNA damage ensue, which is important for tumor progression⁴². The interaction of these aberrant signaling pathways might accelerate metastasis, recurrence, and death in patients of SC. The following descriptions and charts were added to the manuscript:

On page14:

Based on the differences in tumor stages of three histological subtypes, the differences

in protein and biological functions between early-stage (stages I and II) and late-stage (stages III and IV) of tumors were further investigated in each histological subtype separately. Supplementary Figure 6A demonstrated the number of tumor progression landmarks for each histological subtype. Notably, there was a low overlap of tumor progression landmarks between histological subtypes, demonstrating their subtyping specificity. We also revealed specific KEGG signaling pathways that promote tumor progression for each histological subtype (Supplementary Figure 6B). In particular, the SC subtype, with the worst survival rate, exhibited abnormalities in multiple signaling pathways, focusing on cellular processes (such as focal adhesion and regulation of actin cytoskeleton) and organismal systems (such as leukocyte transendothelial migration, FcγR-mediated phagocytosis, and chemokine signaling pathway) (Supplementary Figure 6B and Supplementary Table 7). Among them, focal adhesion promotes tumor development and metastasis through effects on cancer cells and stromal cells of the tumor microenvironment³⁷. The interaction of these aberrant signaling pathways may accelerate metastasis, recurrence, and even death in the SC population.

Supplementary Figure 6. Comparison and characterization of tumor progression-related proteins in three histological subtypes. A. Venn diagram shows the overlap of differential tumor progression landmarks reported in different subtypes of EOC. **B.** KEGG signaling pathways enriched for tumor progression landmarks in the three histological subtypes, respectively.

Q4. In Supplementary Table 1 – please define what “Age” means here – is this the mean or median age of the patients? And SD?

REPLY: We thank the reviewer’s suggestion. The details of the clinicopathological

characteristics of EOC patients in Supplementary Table 3 are not clearly annotated. Age is the age of the patient at the time of diagnosis of ovarian cancer, and is expressed as mean \pm standard deviation (Mean \pm SD). The improved Supplementary Table 3 is as follows:

Supplementary Table 3. Clinicopathological characteristics of epithelial ovarian cancer patients.

Characteristics	Total population	Clear cell carcinoma	Endometrioid carcinoma	Serous carcinoma
No. of patients	239	80	79	80
Mean (SD) age at diagnosis (years)	53.26 \pm 9.70	51.18 \pm 10.15	51.15 \pm 8.77	57.41 \pm 8.83
FIGO Stage				
I	81 (33.89)	55 (68.75)	23 (29.11)	3 (3.75)
II	47 (19.67)	8 (10.00)	28 (35.44)	11 (13.75)
III	88 (36.82)	15 (18.75)	22 (27.85)	51 (63.75)
IV	23 (9.62)	2 (2.50)	6 (7.60)	15 (18.75)
Vital status				
Dead	105 (43.93)	27 (33.75)	22 (27.85)	56 (70.00)
Alive	134 (56.07)	53 (66.25)	57 (72.15)	24 (30.00)
Relapse status				
Yes	151 (63.18)	37 (46.25)	38 (48.10)	76 (95.00)
No	88 (36.82)	43 (53.75)	41 (51.90)	4 (5.00)

Values are numbers (percentages) unless stated otherwise.

Q5. The median age of ovarian cancer, in general, is 63 – while in this cohort of patients is 52, which is 11 years younger than the expected age. Although clear cell OC, on average a few years younger, this difference is also seen in the serous cohort. Could you please explain what could be driving the ovarian cancer diagnosis at such a young age in this cohort of patients?

REPLY: We thank the reviewer for this careful consideration. Globally, the majority of ovarian cancer is most commonly diagnosed after menopause between the ages of 60 to 64 years, with the typical age of 63 years at diagnosis (Cancer Research UK). However, the median age in this cohort was 52 years, and similar phenomenon has been observed in other multicenter clinical studies ^{14,43}. For instance, data were collected from four primarily national cancer centers in China have reported that the median age at diagnosis of ovarian cancer was 55.2 and 53.1 years in the surgical and non-surgical groups, respectively ⁴³. This may depend on ethnic and geographical differences. In

addition, pregnancy history have been consistently related to ovarian cancer risk³⁵. As the fertility rates in China have continued to change in recent years³⁶, this may contribute to the early development of ovarian cancer. It also should be noted that FIGO stage I accounted for a higher proportion (33.89%) of cases in the present study compared with other studies. This phenomenon might be attributed to the accidental diagnosis of EOC when checking for other diseases such as benign ovarian tumors as well as myoma. Furthermore, the Department of Obstetrics and Gynecology of China Medical University is the highest authority on EOC diagnosis in northeast China, potentially explaining the early diagnosis of EOC in the present study⁴⁴.

On page16:

In clinical characteristics, we found that women in our cohort tended to be younger and had a more favorable outcome compared to the general EOC population, which is consistent with several previous studies^{8,12,13,30-33}. In terms of age, previous evidence has shown that pregnancy history has been consistently related to OC risk³⁵. As the fertility rates in China have continued to change in recent years³⁶, this may partly contribute to the early development of OC. Younger age at diagnosis, and thus, better physical fitness and higher tolerance to aggressive treatments, may be another possible explanation for this disparity^{30,31}.

Q6. Please provide survival data in months rather than days. What do these “time” numbers mean in the table OS and RFS? As in almost all the cohorts, less than half of the patients recurred/died – this could not mean median RSF/OS.

REPLY: We thank the reviewer for this careful consideration. We have converted survival time from days to months, and updated the relevant charts, such as survival curves for patients with distinct histological subtypes (Figure 1E). In addition, Figure 1E showed a median OS of 47.57 ± 2.36 months and a median RFS of 16.57 ± 2.50 months in serous carcinoma patients (median \pm standard error). We revised the detailed information of OS (number of alive and dead in vital status) and RFS (number of non-relapsed and relapsed in the relapse status), and the improved Supplementary Table 3 is as follows:

On Page9:

To better depict the inter-tumor heterogeneity of EOC, we characterized the patients' clinical information among pathology subtypes. Compared with EC and CCC, patients

with SC were older at diagnosis and were prone to relapse (Figure 1C and Supplementary Table 3). Moreover, SC was enriched with advanced tumor stages than EC and CCC (Figure 1D and Supplementary Table 3). **In particular, evaluation of the survival characteristics of the EOC histological subtypes revealed that SC, EC and CCC exhibited significantly different OS and RFS (Figure 1E and Supplementary Table 3). Of these, SC had a significantly lower survival rate and a greater risk of postoperative death and recurrence, with a median OS of 47.57 ± 2.36 months and a median RFS of 16.57 ± 2.50 months (median \pm standard error).**

Figure 1. Proteome landscape in EOC histological subtypes. **A.** Overview of the experimental setup for MS-based proteome profiling. **B.** UMAP plot of EOC tumor and CT samples, color-coded by EOC histological subtypes. **C.** Heatmap showing the clinical information and mean OS protein abundance of samples. **D.** Differences in the abundance of EOC histological subtypes in terms of tumor stage. P-values were

calculated by Fisher's exact test. **E.** Kaplan–Meier plots of OS and RFS for EOC histological subtypes. **F.** Schematic diagram of protein subcellular localization.

Supplementary Table 3. Clinicopathological characteristics of epithelial ovarian cancer patients.

Characteristics	Total population	Clear cell carcinoma	Endometrioid carcinoma	Serous carcinoma
No. of patients	239	80	79	80
Mean (SD) age at diagnosis (years)	53.26 ± 9.70	51.18 ± 10.15	51.15 ± 8.77	57.41 ± 8.83
FIGO Stage				
I	81 (33.89)	55 (68.75)	23 (29.11)	3 (3.75)
II	47 (19.67)	8 (10.00)	28 (35.44)	11 (13.75)
III	88 (36.82)	15 (18.75)	22 (27.85)	51 (63.75)
IV	23 (9.62)	2 (2.50)	6 (7.60)	15 (18.75)
Vital status				
Dead	105 (43.93)	27 (33.75)	22 (27.85)	56 (70.00)
Alive	134 (56.07)	53 (66.25)	57 (72.15)	24 (30.00)
Relapse status				
Yes	151 (63.18)	37 (46.25)	38 (48.10)	76 (95.00)
No	88 (36.82)	43 (53.75)	41 (51.90)	4 (5.00)

Values are numbers (percentages) unless stated otherwise.

Q7. Do the authors have information about genetic testing on these patients?

REPLY: We thank the reviewer's suggestion. Due to sample and cost constraints, we only obtained protein expression profiles by sequencing, but did not sequence the genomics of these samples. For this reason, we have also revised the title from "proteogenomics" to "proteomic". With the development of mass spectrometry-based proteomic, it is now possible to characterize disease phenotypes and their regulation by biologically active molecules with unprecedented resolution and dimension⁴⁵. In future work, we will integrate transcriptomic, mutational and copy number variation data to explore the molecular characteristics and therapeutic targets of different histological subtypes in EOC.

On Page19:

In addition, we only obtained mass spectrometry-based proteomic data, lacking data on the transcriptome, mutations, and copy number variation data. This limits our study to proteomic, and multi-omics data will be used to deepen the study of EOC histological subtypes in the future.

Q8. Could you please provide comments on subsequent treatments in recurrent disease.

REPLY: We thank the reviewer for this careful consideration. In our study, CDK4, CDK6, and CDKN1B participating in the cell cycle were all risk factors for the recurrence of epithelial ovarian cancer. Inhibition of CDKs blocks uncontrolled cell proliferation, and the development of pharmacological inhibitors of CDKs has shown promising activity and clinical success in the treatment of breast cancer^{46,47}. Small-molecule CDK4/6-inhibitors (palbociclib, ribociclib, and abemaciclib) are now being tested in over 300 active or recruiting clinical trials for over 50 tumor types, such as ovarian, lung, liver, uterine, and colon cancers⁴⁸. In particular, the effectiveness of the palbociclib was explored in a phase II trial in heavily pretreated ovarian cancer patients⁴⁹. Additionally, EOC cells spread through direct extension to the peritoneum, invade the underlying basement membrane and spread over the ECM to form metastatic implants, unlike most solid tumors that spread by lymphatic or hematogenous routes⁵⁰. Collagen is the most abundant component in ECM proteins, which play a critical role in cell proliferation, differentiation and maintenance of tissue homeostasis⁵¹. Among them, abnormal expression of COL4A1 and COL4A2 disrupts the strict regulation of the ECM and promotes the proliferation and invasion of cancer cells, which is often the main cause of cancer metastasis, recurrence and even death^{52,53}. This evidence suggests that ECM proteins could be novel diagnostic markers for predicting EOC recurrence and promising drug targets for EOC treatment.

On Page17:

Particularly, disturbances in signaling pathways may suggest potential therapeutic opportunities. Uncontrolled cell proliferation is a hallmark of cancer⁵⁴. The complex composed of cyclins and their associated cyclin-dependent kinases (CDKs) promotes cell cycle progression by phosphorylating and inactivating the retinoblastoma protein (RB)^{46,55}. **We observed that CDK4, CDK6, and CDKN1B participating in the cell cycle were all risk factors for the recurrence of EOC. Inhibition of CDKs blocks uncontrolled cell proliferation. The development of pharmacological inhibitors of CDKs has shown promising activity and clinical efficacy in the treatment of breast cancer^{46,56-58}. There was also a phase II trial found that CDK4/6 inhibition with palbociclib was well tolerated and demonstrated single-agent activity in patients with recurrent ovarian cancer⁴⁹.** Additionally, EOC cells spread through direct extension to the peritoneum, invade the underlying basement membrane and spread over the ECM to form metastatic

implants, unlike most solid tumors that spread by lymphatic or hematogenous routes ⁵⁰. Collagen is the most abundant component in ECM proteins, which plays a critical role in cell proliferation, differentiation and maintenance of tissue homeostasis ⁵¹. Among them, abnormal expression of COL4A1 and COL4A2 disrupts the strict regulation of the ECM and promotes the proliferation and invasion of cancer cells, which is often the main cause of cancer metastasis, recurrence and even death ^{52,53}. This evidence suggests that ECM proteins could be novel diagnostic markers for predicting EOC recurrence and promising drug targets for EOC treatment.

Q9. Please comment on the control ovarian samples – what was the age of those patients, and how were these specimens obtained?

REPLY: We thank the reviewer's suggestion. The normal ovarian tissues (n = 30) were taken from cases of uterine fibroids, in which the ovarian was surgically removed incidental to radical surgery for non-ovarian diseases. The median age of women at the time of surgery was 55 years old and ranged between 41 and 71 years old.

On Page4:

Based on the characteristics of EOC, it is extremely difficult to obtain para-carcinoma tissue ¹. Therefore, the histologically normal ovarian tissues (n = 30) taken from cases of uterine fibroids were used as CT samples, in which the ovary was surgically removed incidental to radical surgery. The median age of CT samples at time of surgery was 55 years old and ranged between 41 and 71 years old.

Q10. Please provide data on race in the manuscript and if most of the patients are of East Asian descent – please acknowledge this in the discussion session as a limitation. East Asian patients do have more favorable clinical outcomes compared to other races – and higher rates of clear cell OC. Thus, some of these findings may not be completely generalizable to a different cohort of patients.

REPLY: We thank the reviewer for this careful consideration. The patients in this study were all from the East Asian population, and the incidence and survival of epithelial ovarian cancer in relation to race and geographic region have been added to the discussion section.

On page18:

We recognize several limitations. First, historical epidemiologic data has suggested that

the incidence and survival rates of OC depends on the ethnicity and geographical area^{9,10}. Since East Asian backgrounds were significantly younger compared to other races and have an earlier stage of OC^{11,12}. This could contribute to the fact that our cohort had favorable clinical outcomes than that of the general patients¹³. However, since our study only included patients from China, the data may not be fully representative of the entire population. This inevitably limits generalization to other populations and introduces the possibility of bias towards particular demographics¹⁴.

Reply to Reviewer #4 (expertise in biostatistics and statistical genomics):

Gong et al. quantified the proteome of 269 epithelial ovarian cancer (EOC) samples. The protein ranking lists were generated from a differently expressed test between EOC and normal adjacent tissues, and between EOC histological subtypes. Functional enrichment analysis and protein network analysis were also conducted to interpret the proteome data. It's an opportunity to generate molecular profiles to gain biological and clinical insights into EOC in addition to the well-known Clinical Proteomic Tumor Analysis Consortium (CPTAC) discoveries. However, there is not enough molecular data generated for an EOC molecular landscape. Data analysis were relatively superficial, and some analysis were also flawed and not correct. Overall, the manuscript did not provide biological insight into the EOC, and the major claimed discoveries including molecular signatures and therapeutics targets were not truly validated.

***Q1.** CPTAC already generated two comprehensive proteomics data. The discovery cohort (Zhang et al., 2016, Cell) included 160 high-grade serous carcinomas (HGSC) and the validation cohort (McDermott et al, 2020, Cell Reports Medicine) generated proteomics data from 83 prospectively collected ovarian HGSC. The question for the current manuscript is what new findings were discovered from the cohort with proteome data only compared with comprehensive proteogenomic data from two CPTAC cohorts. Can the major findings be validated in CPTAC cohorts? If not, what's the explanation? The authors should address the main motivations of this study. Why is it different from CPTAC studies? Actually, the CPTAC discovery study (Zhang et al., 2016, Cell) was not even mentioned in this manuscript. There was no clinically meaningful focus in the manuscript. For example, after CPTAC studies, the research communities are very*

interested in the new biomarkers and therapeutic targets for chemotherapy resistance of EOC. However, the current study is still focusing on differently expressed protein list generation.

REPLY: We thank the reviewer for this careful consideration. The main objective of our study was to characterize the molecular signatures and therapeutic targets of different histological subtypes in epithelial ovarian cancer based on proteomic profiles. For this purpose, we first collected three histological subtypes (CCC, EC, and SC) of epithelial ovarian cancer samples. In particular, we focused on the differences in protein expression levels, involvement in biological functions, and prognostic ability among the three histological subtypes.

Previous studies by Zhang et al.⁷ and McDermott et al.¹⁶ profound implications for profiling the proteogenomic characterization of high-grade serous ovarian cancer (HGSOC), particularly with regard to the detailed analysis of the molecular components, potential mechanisms, and therapeutic targets associated with HGSOC. Our study is consistent with previous studies in terms of protein expression abundance, biological function, and prognostic ability. At the protein expression level, we also found that most of the dysregulated proteins tended to be down-regulated in tumor tissue, such as proteins involved in ECM-receptor interaction (COL1A1, COL1A2, COL2A1, COL4A1, COL4A2, COL6A1, COL6A2, COL6A3, LAMA2, LAMA4, LAMB2, etc.), and proteins involved in PI3K-AKT signaling pathways (ITGA2B, ITGB3, etc.)¹⁶. At the functional level, we have found that the dysregulated proteins are significantly enriched in biological processes such as humoral immune response, focal adhesion, and regulation of endocytosis¹⁶. At the prognostic value level, high expression of COL1A2 and IFIT3 was found to be associated with poor patient prognosis in our study, which was also validated in CPTAC cohort. These consistent results significantly increase the credibility of our study.

In addition, our study highlights the comparison of the three histological subtypes (CCC, EC, and SC) in terms of proteomic characteristics, whereas previous studies have focused on high-grade SC. Firstly, we found that SC, EC, and CCC exhibited significantly different OS and RFS, with SC showing significantly lower survival and a greater risk of postoperative death and recurrence (Figure 1E). Secondly, we found multiple co-expression modules with different regulatory roles in the three histological subtypes. For example, the protein of module 31 was not only differentially expressed between different histological subtypes (Figure 4C), but also was highly expressed at

advanced pathological stages, suggesting that IFIT3 may be a tumor progression landmark (Figure 4E-F). Then, we also identified histological subtype-specific proteins and their participation in biological processes or signaling pathways (Figure 5). Finally, we screened potential therapeutic targets for different histological subtypes of EOC based on protein expression levels, prognostic ability and druggability. In particular, the MPP7 protein showed good potential as a therapeutic target for SC, with subtype-specific high expression and prognostic ability. The loss-of-function experiments also demonstrated the involvement of MPP7 in regulation of cell proliferation, cell cycle progression, cell apoptosis, cell migration, and cell invasion in SC in vitro, which highlighted the importance of MPP7 in SC. In conclusion, we provide new insights based on proteomic characterization to explore the molecular signatures and therapeutic targets in distinct histological subtypes of EOC.

The description and comparison of previous studies have been added to the Introduction and Discussion sections of the manuscript, respectively.

On Page3:

Furthermore, several studies have explored the important role of proteomic profiling in high-grade serous ovarian cancer (HGSOC). The study by Zhang et al. provided additional insights into the pathways and processes that drive the biology of HGSOC and how these pathways are altered relative to the clinical phenotypes ⁷. And, McDermott et al. described a potential role for proliferation-induced replication stress in promoting the characteristic chromosomal instability of HGSOC ¹⁶. A study from Coscia et al. also revealed that cancer/testis antigen 45 (CT45), a prognostic factor associated with the doubling of disease-free survival, enhanced chemosensitivity in metastatic HGSOC ¹⁷. However, molecular characterization of EOC histological subtypes (CCC, EC, and SC) using large cohorts is still limited.

On Page17:

Notably, most dysregulated proteins tended to be under-expressed in tumor tissues, which was consistent with previous study ¹⁶. For example, the expression abundance of multiple key proteins (such as COL1A1, COL1A2, ITGA2B, ITGB3, etc.) in the ECM-receptor interaction or PI3K-AKT signaling pathways was significantly down-regulated ¹⁶. And, significant enrichment of biological functions, such as humoral immune response, focal adhesion, and regulation of endocytosis, was corroborated ¹⁶.

Q2. A lot of crucial genomic information was missing for this “proteogenomic” study.

The major finding and conclusions of this manuscript were basically a differential protein expression list. For a more comprehensive study, the following genomic data need to be generated:

(a) Mutation and CNV data. It may be difficult to generate whole exome sequencing data, but targeted sequencing for EOC common mutations/CNVs is necessary.

(b) Transcriptome data. Technically it's not difficult to generate the whole transcriptome data from FFPE samples, and it's very important to integrate the transcriptome data and proteome data to understand the molecular changes of EOC, and to reduce possible false discoveries.

(c) Ideally, both phosphoproteome and acetylome data, or at the least phosphoproteome data need to be generated. The RNA, protein, and phosphorite levels data are needed to describe changes in important oncogenic signaling pathways, in order to understand the biology underlying recurrence and treatment resistance.

Overall very limited data (and analysis) was added by this study as a proteogenomic characterization paper, and it is quite difficult to generate biological hypotheses from proteome data only.

REPLY: We thank the reviewer's suggestion. In combination with recent developments in mass spectrometry-based proteomic instrumentation and data analysis pipelines, have now enabled the dissection of disease phenotypes and their modulation by bioactive molecules at unprecedented resolution and dimensionality ⁴⁵. Our study focused on proteomic explored the molecular features and therapeutic targets of the different histological subtypes in EOC, and validated the relevant signatures using external datasets, PRM analysis, and basic experiments. For reasons of sample and cost, we did not obtain multi-omics data on transcriptome, mutation, CNV, phosphoproteome and acetylation, but only obtained protein expression profiles of EOC patients through sequencing. For this reason, we have also revised the title from "proteogenomics" to "proteomic". In future work, we will perform more detailed multi-omics analysis of the markers identified in our study in order to explore their important role in EOC.

On Page19:

In addition, we only obtained mass spectrometry-based proteomic data, lacking data on the transcriptome, mutations, and copy number variation data. This limits our study to proteomic, and multi-omics data will be used to deepen the study of EOC histological subtypes in the future.

Q3. There was no multiple comparison adjustment for several important analysis. For example, the functional enrichment analysis on Page 7, the identification of differentially expressed proteins among three histological subtypes on Page 6, and the survival analysis report in SupTable 3. For high-dimensional genomic/proteomic data, it is easy to have false positive findings. The adjusted p-values should be used instead of nominal p-values.

REPLY: We thank the reviewer for this careful consideration. After careful checking, differential expression analysis (whether protein abundance differed between cancer and control groups, or between the three histological subtypes), functional enrichment analysis, and survival analysis in the study were corrected by the by multiple hypothesis testing. Relevant details have been corrected in our manuscript.

On Page6:

2.4 Differential Expression Analysis of the Proteome

Differential proteomic analysis was conducted on 4,447 quantifiable proteins in a total of 8,257 proteins detected. We performed Wilcoxon signed-rank tests to identify the dysregulated proteins with a statistically significant P-value between EOC and CT patients. The P-values were corrected by the **Benjamini-Hochberg (BH) procedure**. Significantly up- or down-regulated proteins were extracted by a threshold of **adj.P-value < 0.01 and |log2 (fold change)| > 1**.

To assess differentially expressed proteins across treatment groups, **Kruskal-Wallis test (K-W test)** was used to identify differentially expressed proteins among the three histological subtypes of EOC (SC, EC, and CCC). Post-hoc tests were performed to identify the differentially expressed proteins between any two subtypes (**adj.P-values < 0.05, R package ‘PMCMRplus’**).

2.5 Survival Analysis

For the clinicopathological analysis, Fisher’s exact test (two-sided) was performed. Kaplan–Meier curves and log-rank tests were used to compare OS or RFS among the proteomic subtypes. Clinical associations of protein expression was examined by the Cox proportional hazards model, **and P-values were corrected by the False Discovery Rate (FDR) procedure**. Univariable and multivariable Cox regressions were applied to estimate the hazard ratios (HR), 95% confidence intervals (CI), **Cox P-values and Cox adj.P-values** of each protein.

2.6 Functional Enrichment Analysis

Comprehensive function annotation of proteins, including Gene Ontology (GO) ^{19,20}, Kyoto Encyclopedia of Genes and Genomes (KEGG) pathways ²¹ and Gene Set Enrichment Analysis (GSEA) ²², was performed on R package ‘clusterProfiler’ ²³, Metascape (<http://www.metascape.org/>) and R package ‘fgsea’ ²⁴ to identify GO biological processes (BPs), KEGG pathways, Reactome gene sets ²⁵, WikiPathways ²⁶, and Hallmark gene sets ^{22,27} in which dysregulated proteins were enriched. The **adj.P-values < 0.05** were considered statistically significant. R packages ‘simplifyEnrichment’ ²⁸ and ‘GOsemSim’ ²⁹ were used to cluster GO terms based on similarity matrices of functional terms. The function ‘simplify’ of R package ‘clusterProfiler’ was used to remove redundancies of enriched GO results.

Q4. The Kolmogorov-Smirnov test on Page 6 may not be correct since the KS test is not for testing multiple groups.

REPLY: We thank the reviewer for this careful consideration. Apologies for the writing mistake, the content of the manuscript has been corrected and "Kolmogorov-Smirnov test" has been amended to "Kruskal-Wallis test". A Kruskal-Wallis test is used to determine whether or not there is a statistically significant difference between the medians of three or more independent groups.

On Page6:

To assess differentially expressed proteins across treatment groups, **Kruskal-Wallis test (K-W test)** was used to identify differentially expressed proteins among the three histological subtypes of EOC (SC, EC, and CCC). Post-hoc tests were performed to identify the differentially expressed proteins between any two subtypes (**adj.P-values < 0.05, R package ‘PMCMRplus’**).

Q5. For the description and analysis of clinical data from three subtypes in 3.1, the results may be biased due to data collection. Are the results consistency with the previous studies? There were no literature reviews and discussions. The clinical data including age, stage, survival outcomes, etc of each individual patient should be included in the supplementary data.

REPLY: We thank the reviewer for this careful consideration. Firstly, we selected EOC samples that included at least 90% tumor cells in our study. All these samples were examined by two experienced pathologists who confirmed the histological subtype (SC, EC, or CCC) to which the sample belonged. These criteria ensured the accuracy of the

sample collection.

Secondly, at the level of clinicopathological characteristics of the EOC samples (e.g., age, prognosis and tumor staging of histological subtypes), our findings are consistent with multiple previous studies. Our study demonstrated that SC patients had a higher proportion of late-stage (stages III and IV) and higher mortality and recurrence rates compared to EC and CCC. The phenomenon is consistent with previously published data on histotype-specific survival patterns⁵⁹⁻⁶³. For example, Peres et al observed that the women with SC had worse outcomes than others using SEER data⁶⁰. Chiang et al investigated the changes of incidence and prognosis of epithelial ovarian cancer in thirty years in Taiwan, their data showed the 10-year survival of patients with EC or CCC was better than those of SC⁶³. In the UK Million Women Study, the majority of EC, CCC, and mucinous carcinomas were diagnosed at stage I+II, but the majority of SC and cases of other/ unspecified histological type were diagnosed at stage III+IV⁵⁹. In addition, women in our cohort tended to be younger, at an earlier stage of diagnosis, and with more favorable prognostic outcomes, which is consistent with some reports assessing survival differences in EOC in Asians. For example, Fuh et al. published results in OC patients between white and Asian women from the SEER program over a 21 year period, demonstrating that Asian patients are younger than white patients⁸. And it was determined that Asian women maintained a survival advantage after adjusting for age, stage, histology, and extent of surgical treatment. This reflected the differences in mortality rates among EOC patients by race/ethnicity.

Then, at the level of the molecular characteristics of the EOC samples, our findings are also consistent with multiple previous studies. For example, most dysregulated proteins tended to be under-expressed in tumor tissues, which was consistent with previous studies¹⁶. ECM-receptor interaction-related proteins (COL1A1, COL1A2, COL2A1, COL4A1, COL4A2, COL6A1, COL6A2, COL6A3, LAMA2, LAMA4, LAMB2, etc.) and PI3K-AKT signaling-related proteins (ITGA2B, ITGB3, etc.) showed significant down-regulation (adj.P-value < 0.01 and $|\log_2(\text{fold change})| > 1$) in both our dataset and the CPTAC cohort¹⁶. And, multiple significantly enriched biological processes or signaling pathways showed a consistent, e.g., humoral immune response, focal adhesion, and regulation of endocytosis. In addition, multiple proteins also showed consistent prognostic value in the two datasets, such as COL1A2 and IFIT3 (log-rank P-value < 0.05).

Finally, a literature review of our study results with previous studies has been added to

the Discussion section. Moreover, the clinicopathological characteristics of EOC patients have been also added to the supplementary materials (Supplementary Table 1).

On Page16:

In clinical characteristics, we found that women in our cohort tended to be younger and had a more favorable outcome compared to the general EOC population, which is consistent with several previous studies^{8,12,13,30-33}. In terms of age, previous evidence has shown that pregnancy history has been consistently related to OC risk³⁵. As the fertility rates in China have continued to change in recent years³⁶, this may partly contribute to the early development of OC. Younger age at diagnosis, and thus, better physical fitness and higher tolerance to aggressive treatments, may be another possible explanation for this disparity^{30,31}.

On Page17:

Notably, most dysregulated proteins tended to be under-expressed in tumor tissues, which was consistent with previous study¹⁶. For example, the expression abundance of multiple key proteins (such as COL1A1, COL1A2, ITGA2B, ITGB3, etc.) in the ECM-receptor interaction or PI3K-AKT signaling pathways was significantly down-regulated¹⁶. And, significant enrichment of biological functions, such as humoral immune response, focal adhesion, and regulation of endocytosis, was corroborated¹⁶.

Q6. The complete protein and pathway analysis ranking list including differentially expressed outcomes, functional enrichment analysis, and survival analysis should be listed in the supplementary data.

REPLY: We thank the reviewer's suggestion. We have added details of differential expression analysis results, functional enrichment analysis results and survival analysis results in the supplementary table (Supplementary Tables 4-6).

Q7. In the result section 3.2, the discovery and validation of diagnostic markers may not be meaningful since it's unlikely to use protein biomarkers for diagnostic purposes.

REPLY: We thank the reviewer for this careful consideration. Our expression "discovery and validation of diagnostic markers" is not accurate and should be revised to "multiple up-regulated proteins have the potential ability to distinguish between EOC and CT samples", and the details in the manuscript have been revised. Cancer exosomal proteomic research is currently in its nascent stage⁶⁴. In future work, we will conduct more detailed analysis of the exosomal proteins identified in this study to explore their

important role in EOC diagnosis.

On page 10:

We aimed to measure the ability of differentially expressed proteins to classify EOC samples and CT samples. Firstly, differentially expressed proteins were used as the initial feature to discriminate tumor samples from normal samples. Next, the area under the curve (AUC) of receiver operating characteristic (ROC) curves obtained from the R package ‘pROC’⁶⁵ was used to evaluate the classification performance of each feature. In addition, proteins with good discriminatory power were enrolled for further validation in the ovarian cancer cohort of McDermott, J. E. et al. (Clinical Proteomic Tumor Analysis Consortium, CPTAC cohort)¹⁶. We observed that multiple up-regulated proteins exhibited strong discriminatory ability with a mean AUC greater than 0.9 (current cohort) and further validated in the CPTAC cohort¹⁶ with a mean AUC also greater than 0.8. We also integrated exosome protein lists from the ExoCarta (<http://www.exocarta.org/>)⁶⁶ and Vesiclepedia (<http://microvesicles.org/>) databases⁶⁷. Figure 2E demonstrated that some exosome proteins had the potential ability to distinguish between EOC samples and CT samples.

Q8. There was no validation for the biomarker discovery from section 3.4.

REPLY: We thank the reviewer’s suggestion. Firstly, we used Parallel Reaction Monitoring (PRM) analysis to verify the expression stability of important proteins in the modules (Module7 and Module31). PRM analysis is an ion monitoring technology based on high-resolution, high-precision mass spectrometry, which can selectively detect target proteins, to achieve Quantification of target proteins. The results demonstrated that the expression levels of IFIT1, IFIT2, IFIT3, COL4A1, COL4A2, and LAMA1 proteins in histological subtypes and pathological stages were consistent with the findings from the LC-MS/MS analysis (Supplementary Figure 4F-G). For example, IFIT1, IFIT2, and IFIT3 proteins in module 31 were not only differentially expressed between different histological subtypes, but were also highly expressed in late pathological stages (stage III and IV). Moreover, ECM receptor members including COL4A1, COL4A2, and LAMA1, which belong to Module7, were also associated with subtypes and stages, consistent with the results of proteomic analysis. These results indicated the expressed stability of these proteins. Further, we validated the prognostic value of IFIT3 proteins using an independent dataset (CPTAC cohort¹⁶). The results supported that high expression of IFIT3 protein was associated with shorter overall

survival of patients (survival analysis by taking the optimal cutpoint, log-rank P-values < 0.01), which is also consistent with the original results.

Supplementary Figure 4. Protein expression of core modules in EOC. A-B. Boxplots illustrating the abundance of Module31 proteins in different histological subtypes and stages. **C.** Sub-network of Module7. **D-E.** Boxplots illustrated the abundance of COL4A1, COL4A2, and LAMA1 in the different histological subtypes and stages. **F-G.** In the PRM analysis, the expression characteristics of these proteins, including IFIT1, IFIT2, IFIT3, COL4A1, COL4A2, and LAMA1, in histological subtypes and pathological stages were verified. **The asterisk character represent the significance of the expression discrepancy, *P-value < 0.05; **P-value < 0.01; ***P-value < 0.001; ****P-value < 0.0001. And, ns represents not significant.**

Q9. There was no solid foundation to claim the potential therapeutic targets in section 3.6. Those proteins are just top ranking candidates from differentially expressed protein

lists and/or survival analysis lists. Nothing was related to the true biological mechanisms. The mechanism experiments including cell lines and animal experiments need to perform to confirm the findings.

REPLY: We thank the reviewer for this careful consideration. Proteomic provides in-depth and quantitative characterization of tumor tissue, as well as bringing new biological and diagnostic knowledge of malignant tumors ¹⁸.

In Result 3.6, we found that MPP7 was specifically highly expressed in SC and was associated with poor prognosis. The prognostic value of MPP7 was also validated in the CPTAC dataset (log-rank P-values < 0.05). Subsequently, we further validated the biological function of MPP7 in SC. We performed CCK-8 assays, flow cytometric analysis of cell cycle and cell apoptosis, and transwell assays. The results showed that shRNA-mediated MPP7 knockdown inhibited cell proliferation, cell cycle progression, cell migration, and cell invasion but promoted cell apoptosis in SC. These results demonstrated the biological function of MPP7 in SC development and metastasis.

On Page15:

Based on the prognostic value of MPP7 and the degree of malignancy of SC, we investigated the function of MPP7 in SC cells in vitro. Malignant behaviors including cell proliferation, cell migration, and cell invasion, cell cycle distribution, and cell apoptosis were assessed. CCK-8 assays showed that shRNA-mediated MPP7 knockdown decreased cell viability (Supplementary Figure 7A and Supplementary Table 8), indicating the inhibition of cell proliferation by MPP7 knockdown. Flow cytometric analysis of cell cycle demonstrated that MPP7 knockdown resulted in decreases in cells at the S phase and increases in cells at the G1 and G2 phase (Figure 6G and Supplementary Table 9). The results implied that MPP7 was involved in regulation of G1-S and S-G2 transition. Flow cytometric analysis of cell apoptosis elucidated that cell apoptosis was induced by MPP7 knockdown (Figure 6H and Supplementary Table 10). Transwell assays suggested that cell migration and invasion were inhibited by MPP7 knockdown (Supplementary Figure 7B&C and Supplementary Table 11&12). These findings uncovered that the biological function of MPP7 in SC. Taken together, we propose further analysis of these subtype-specific proteins as promising therapeutic targets in the three histological subtypes.

Figure 6. Potential therapeutic targets for distinct histological subtypes.

A, C, E. Expression levels of CSPG4, TMEM87A, and MPP7 among the respective histological subtypes. **B, D, F.** Kaplan-Meier survival curves for patients expressing CSPG4, TMEM87A, and MPP7 in CCC, EC, and SC patients, respectively. Among them, the red line represents highly expressed protein and the blue line represents lowly expressed protein. **G.** OVCAR-3, A2780, and ES-2 cells were infected with lentiviral vectors carrying shMPP7 or shNC. Cell cycle distribution was analyzed by flow cytometry at 48 h after infection. The gating strategy was shown in Supplementary

Figure 8A. **H.** Cell apoptosis was measured by Annexin V-FITC/PI staining at 48 h after infection. The count of Annexin V-FITC-positive and PI-positive cells (Late apoptosis) and Annexin V-FITC-positive and PI-negative cells (Early apoptosis) was assessed by flow cytometry. The gating strategy was shown in Supplementary Figure 8B. **P-value < 0.01 and **** P-value < 0.0001 vs. the shNC group (n=4): data were analyzed by the one-way analysis of variance (ANOVA) followed by Tukey's tests or the Brown-Forsythe and Welch ANOVA tests followed by Tamhane T2 tests.

Supplementary Figure 7. Effect of MPP7 knockdown on the malignant behavior of epithelial ovarian cancer cells. OVCAR-3, A2780, and ES-2 cells were infected with lentiviral vectors carrying shMPP7 or shNC. **A.** CCK-8 assays were performed at 0, 24, 48, 72, and 96 h after infection. **B-C.** Cell migration (B) and invasion (C) were determined by transwell assays at 48 h after infection. Scale bar: 100 μ m. *P-value < 0.05, ** P-value < 0.01, and **** P-value < 0.0001 vs. the shNC group (n=4): data were analyzed by the one-way or two-way analysis of variance (ANOVA) followed by Tukey's tests.

Supplementary Figure 8. Gating strategies for flow cytometric analysis. **A.** Gating strategies for cell cycle analysis. The major cell population was selected based on FSC and SSC, and then the single cells were gated by PI-A and PI-H. The signal of PI in the single-cell population was analyzed. **B.** Gating strategies for cell apoptosis analysis (Annexin V-FITC-PI staining). Cells were examined by FSC and SSC to obtain the major cell population. Then, the signal of FITC and PI in the cell population was analyzed.

On Page8:

2.9 Experimental methods and statistical analysis

Cell culture and infection Ovarian cancer cell lines OVCAR-3, A2780, and ES-2 were cultured in RPMI-1640 medium (Solarbio, China) containing 20% FBS (TIANHANG, China), RPMI-1640 medium containing 10% FBS, and McCOY's 5A medium (Procel, China) containing 10% FBS at 37°C with 5% CO₂, respectively. Cells were infected with lentiviral vectors carrying short hairpin RNA (shRNA) targeting MPP7 (shMPP7) or its negative control shRNA (shNC) (OVCAR-3: MOI = 30; A2780: MOI = 10; ES-2: MOI = 5).

Cell counting kit-8 (CCK-8) assay Cells (3×10^3) were seeded onto a 96-well plate and harvested at 0, 24, 48, 72, or 96 h after infection. The CCK-8 assays were performed using the CCK-8 assay kit (Wanleibio, China) according to the manufacturer's instructions. The optical density (OD) was detected at 450 nm using a BioTek 800 TS plate reader (Agilent, USA).

Flow cytometry Cell cycle distribution and cell apoptosis were analyzed using flow cytometry. For cell cycle analysis, cells were fixed with 70% cold ethanol and washed with PBS at 48 h after infection. After incubation with RNase A at 37°C for 30 min, the

cells were stained with Propidium Iodide (PI) for 30 min. For cell apoptosis detection, cells were washed with PBS at 48 h after infection and resuspended in binding buffer. Then, the cells were stained with Annexin V-FITC and PI for 15 min. The Cell Cycle Analysis Kit and the Annexin-FITC-PI Staining Kit were purchased from Wanleibio (China). The signal of PI and Annexin V-FITC was detected by a NovoCyte flow cytometer (Agilent, USA). The results obtained from flow cytometric analysis were analyzed by the NovoExpress software (version 1.4.1, Agilent, USA).

Transwell assay The transwell chambers used for migration assays or the matrigel-coated transwell chambers used for invasion assays were placed into 24-well plates. Cells were resuspended in serum-free medium at 48 h after infection and seeded in the transwell upper chamber at 5,000 (Migration assay) or 50,000 (Invasion assay) cells per well. After 24 h, the cells on the lower side of the transwell membrane were fixed with 4% PFA (Aladdin) for 20 min, then stained with 0.5% crystal violet (Amresco, USA) for 5 min, and counted under a microscope (OLYMPUS, Japan).

Statistical analysis GraphPad Prism 8 (GraphPad Software, USA) was used for statistical analyses. The Shapiro-Wilk normality test and the Brown-Forsythe test were used for analysis of normal distribution and variance homogeneity, respectively. Data, which were normally distributed and had equal variances, were analyzed using one-way or two-way analysis of variance (ANOVA) followed by Tukey's tests. Data, which were normally distributed and had unequal variances, were analyzed using Brown-Forsythe and Welch ANOVA tests followed by Tamhane T2 tests. Data are presented as mean \pm standard deviation (SD). A p value < 0.05 was considered statistically significant.

Eventually, we would like to thank the reviewer again for taking the time to review our manuscript.

References

- 1 Cai, M. *et al.* Expression of hMOF in different ovarian tissues and its effects on ovarian cancer prognosis. *Oncol Rep* **33**, 685-692, doi:10.3892/or.2014.3649 (2015).
- 2 Harel, M. *et al.* Proteomics of Melanoma Response to Immunotherapy Reveals Mitochondrial Dependence. *Cell* **179**, 236-250 e218, doi:10.1016/j.cell.2019.08.012 (2019).
- 3 Nie, X. *et al.* Multi-organ proteomic landscape of COVID-19 autopsies. *Cell* **184**, 775-791 e714, doi:10.1016/j.cell.2021.01.004 (2021).
- 4 Jiang, Y. *et al.* Proteomics identifies new therapeutic targets of early-stage hepatocellular

- carcinoma. *Nature* **567**, 257-261, doi:10.1038/s41586-019-0987-8 (2019).
- 5 Ma, W. *et al.* DreamAI: algorithm for the imputation of proteomics data. 2020.2007.2021.214205, doi:10.1101/2020.07.21.214205 %J bioRxiv (2021).
- 6 Ritchie, M. E. *et al.* limma powers differential expression analyses for RNA-sequencing and microarray studies. *Nucleic Acids Res* **43**, e47, doi:10.1093/nar/gkv007 (2015).
- 7 Zhang, H. *et al.* Integrated Proteogenomic Characterization of Human High-Grade Serous Ovarian Cancer. *Cell* **166**, 755-765, doi:10.1016/j.cell.2016.05.069 (2016).
- 8 Fuh, K. C. *et al.* Survival differences of Asian and Caucasian epithelial ovarian cancer patients in the United States. *Gynecol Oncol* **136**, 491-497, doi:10.1016/j.ygyno.2014.10.009 (2015).
- 9 Matulonis, U. A. *et al.* Ovarian cancer. *Nat Rev Dis Primers* **2**, 16061, doi:10.1038/nrdp.2016.61 (2016).
- 10 Torre, L. A. *et al.* Ovarian cancer statistics, 2018. *CA Cancer J Clin* **68**, 284-296, doi:10.3322/caac.21456 (2018).
- 11 Sung, P. L. *et al.* Global distribution pattern of histological subtypes of epithelial ovarian cancer: a database analysis and systematic review. *Gynecol Oncol* **133**, 147-154, doi:10.1016/j.ygyno.2014.02.016 (2014).
- 12 Peres, L. C. & Schildkraut, J. M. Racial/ethnic disparities in ovarian cancer research. *Adv Cancer Res* **146**, 1-21, doi:10.1016/bs.acr.2020.01.002 (2020).
- 13 Cowan, R. A. *et al.* Exploring the impact of income and race on survival for women with advanced ovarian cancer undergoing primary debulking surgery at a high-volume center. *Gynecol Oncol* **149**, 43-48, doi:10.1016/j.ygyno.2017.11.012 (2018).
- 14 Gao, Q. *et al.* Olaparib Maintenance Monotherapy in Asian Patients with Platinum-Sensitive Relapsed Ovarian Cancer: Phase III Trial (L-MOCA). *Clin Cancer Res* **28**, 2278-2285, doi:10.1158/1078-0432.CCR-21-3023 (2022).
- 15 Asleh, K. *et al.* Proteomic analysis of archival breast cancer clinical specimens identifies biological subtypes with distinct survival outcomes. *Nat Commun* **13**, 896, doi:10.1038/s41467-022-28524-0 (2022).
- 16 McDermott, J. E. *et al.* Proteogenomic Characterization of Ovarian HGSC Implicates Mitotic Kinases, Replication Stress in Observed Chromosomal Instability. *Cell Rep Med* **1**, doi:10.1016/j.xcrm.2020.100004 (2020).
- 17 Coscia, F. *et al.* Multi-level Proteomics Identifies CT45 as a Chemosensitivity Mediator and Immunotherapy Target in Ovarian Cancer. *Cell* **175**, 159-170 e116, doi:10.1016/j.cell.2018.08.065 (2018).
- 18 Mani, D. R. *et al.* Cancer proteogenomics: current impact and future prospects. *Nat Rev Cancer*, doi:10.1038/s41568-022-00446-5 (2022).
- 19 Ashburner, M. *et al.* Gene ontology: tool for the unification of biology. The Gene Ontology Consortium. *Nat Genet* **25**, 25-29, doi:10.1038/75556 (2000).
- 20 Gene Ontology, C. The Gene Ontology resource: enriching a Gold mine. *Nucleic Acids Res* **49**, D325-D334, doi:10.1093/nar/gkaa1113 (2021).
- 21 Kanehisa, M., Furumichi, M., Sato, Y., Ishiguro-Watanabe, M. & Tanabe, M. KEGG: integrating viruses and cellular organisms. *Nucleic Acids Res* **49**, D545-D551, doi:10.1093/nar/gkaa970 (2021).
- 22 Subramanian, A. *et al.* Gene set enrichment analysis: a knowledge-based approach for interpreting genome-wide expression profiles. *Proc Natl Acad Sci U S A* **102**, 15545-15550,

- doi:10.1073/pnas.0506580102 (2005).
- 23 Wu, T. *et al.* clusterProfiler 4.0: A universal enrichment tool for interpreting omics data. *Innovation (N Y)* **2**, 100141, doi:10.1016/j.xinn.2021.100141 (2021).
- 24 Korotkevich, G. *et al.* Fast gene set enrichment analysis. 060012, doi:10.1101/060012 %J bioRxiv (2021).
- 25 Gillespie, M. *et al.* The reactome pathway knowledgebase 2022. *Nucleic Acids Res* **50**, D687-D692, doi:10.1093/nar/gkab1028 (2022).
- 26 Martens, M. *et al.* WikiPathways: connecting communities. *Nucleic Acids Res* **49**, D613-D621, doi:10.1093/nar/gkaa1024 (2021).
- 27 Liberzon, A. *et al.* The Molecular Signatures Database (MSigDB) hallmark gene set collection. *Cell Syst* **1**, 417-425, doi:10.1016/j.cels.2015.12.004 (2015).
- 28 Gu, Z. & HΓObschmann, D. simplifyEnrichment: an R/Bioconductor package for Clustering and Visualizing Functional Enrichment Results. 2020.2010.2027.312116, doi:10.1101/2020.10.27.312116 %J bioRxiv (2021).
- 29 Yu, G. *et al.* GOSemSim: an R package for measuring semantic similarity among GO terms and gene products. *Bioinformatics* **26**, 976-978, doi:10.1093/bioinformatics/btq064 (2010).
- 30 Cress, R. D., Chen, Y. S., Morris, C. R., Petersen, M. & Leiserowitz, G. S. Characteristics of Long-Term Survivors of Epithelial Ovarian Cancer. *Obstet Gynecol* **126**, 491-497, doi:10.1097/AOG.0000000000000981 (2015).
- 31 Wang, F. *et al.* Racial/Ethnic Disparities in Mortality Related to Access to Care for Major Cancers in the United States. *Cancers (Basel)* **14**, doi:10.3390/cancers14143390 (2022).
- 32 Cabasag, C. J. *et al.* Ovarian cancer today and tomorrow: A global assessment by world region and Human Development Index using GLOBOCAN 2020. *Int J Cancer* **151**, 1535-1541, doi:10.1002/ijc.34002 (2022).
- 33 Geiger, M. *et al.* Complex formation between urokinase and plasma protein C inhibitor in vitro and in vivo. *Blood* **74**, 722-728 (1989).
- 34 Phan, V. H. *et al.* Ethnic differences in drug metabolism and toxicity from chemotherapy. *Expert Opin Drug Metab Toxicol* **5**, 243-257, doi:10.1517/17425250902800153 (2009).
- 35 Husby, A., Wohlfahrt, J. & Melbye, M. Pregnancy duration and ovarian cancer risk: A 50-year nationwide cohort study. *Int J Cancer* **151**, 1717-1725, doi:10.1002/ijc.34192 (2022).
- 36 Vollset, S. E. *et al.* Fertility, mortality, migration, and population scenarios for 195 countries and territories from 2017 to 2100: a forecasting analysis for the Global Burden of Disease Study. *Lancet* **396**, 1285-1306, doi:10.1016/S0140-6736(20)30677-2 (2020).
- 37 Sulzmaier, F. J., Jean, C. & Schlaepfer, D. D. FAK in cancer: mechanistic findings and clinical applications. *Nat Rev Cancer* **14**, 598-610, doi:10.1038/nrc3792 (2014).
- 38 van Buul, J. D. & Hordijk, P. L. Signaling in leukocyte transendothelial migration. *Arterioscler Thromb Vasc Biol* **24**, 824-833, doi:10.1161/01.ATV.0000122854.76267.5c (2004).
- 39 Cook-Mills, J. M. & Deem, T. L. Active participation of endothelial cells in inflammation. *J Leukoc Biol* **77**, 487-495, doi:10.1189/jlb.0904554 (2005).
- 40 Bartek, J., Bartkova, J. & Lukas, J. DNA damage signalling guards against activated oncogenes and tumour progression. *Oncogene* **26**, 7773-7779, doi:10.1038/sj.onc.1210881 (2007).
- 41 Huang, T. T., Lampert, E. J., Coats, C. & Lee, J. M. Targeting the PI3K pathway and DNA damage response as a therapeutic strategy in ovarian cancer. *Cancer Treat Rev* **86**, 102021, doi:10.1016/j.ctrv.2020.102021 (2020).

- 42 Hills, S. A. & Diffley, J. F. DNA replication and oncogene-induced replicative stress. *Curr Biol* **24**, R435-444, doi:10.1016/j.cub.2014.04.012 (2014).
- 43 Shi, T. *et al.* Secondary cytoreduction followed by chemotherapy versus chemotherapy alone in platinum-sensitive relapsed ovarian cancer (SOC-1): a multicentre, open-label, randomised, phase 3 trial. *Lancet Oncol* **22**, 439-449, doi:10.1016/S1470-2045(21)00006-1 (2021).
- 44 Wang, L. *et al.* ABO Blood Type Has No Impact on Survival in Patients with Epithelial Ovarian Cancer. *J Cancer* **9**, 4334-4340, doi:10.7150/jca.27734 (2018).
- 45 Meissner, F., Geddes-McAlister, J., Mann, M. & Bantscheff, M. The emerging role of mass spectrometry-based proteomics in drug discovery. *Nat Rev Drug Discov* **21**, 637-654, doi:10.1038/s41573-022-00409-3 (2022).
- 46 Goel, S., DeCristo, M. J., McAllister, S. S. & Zhao, J. J. CDK4/6 Inhibition in Cancer: Beyond Cell Cycle Arrest. *Trends Cell Biol* **28**, 911-925, doi:10.1016/j.tcb.2018.07.002 (2018).
- 47 Goel, S., Bergholz, J. S. & Zhao, J. J. Targeting CDK4 and CDK6 in cancer. *Nat Rev Cancer* **22**, 356-372, doi:10.1038/s41568-022-00456-3 (2022).
- 48 Fassl, A., Geng, Y. & Sicinski, P. CDK4 and CDK6 kinases: From basic science to cancer therapy. *Science* **375**, eabc1495, doi:10.1126/science.abc1495 (2022).
- 49 Konecny, G. E. *et al.* A multicenter open-label phase II study of the efficacy and safety of palbociclib a cyclin-dependent kinases 4 and 6 inhibitor in patients with recurrent ovarian cancer. **34**, 5557-5557, doi:10.1200/JCO.2016.34.15_suppl.5557 (2016).
- 50 Iwanicki, M. P. *et al.* Ovarian cancer spheroids use myosin-generated force to clear the mesothelium. *Cancer Discov* **1**, 144-157, doi:10.1158/2159-8274.CD-11-0010 (2011).
- 51 Biteau, B., Hochmuth, C. E. & Jasper, H. Maintaining tissue homeostasis: dynamic control of somatic stem cell activity. *Cell Stem Cell* **9**, 402-411, doi:10.1016/j.stem.2011.10.004 (2011).
- 52 Mohan, V., Das, A. & Sagi, I. Emerging roles of ECM remodeling processes in cancer. *Semin Cancer Biol* **62**, 192-200, doi:10.1016/j.semcancer.2019.09.004 (2020).
- 53 Chaffer, C. L. & Weinberg, R. A. A perspective on cancer cell metastasis. *Science* **331**, 1559-1564, doi:10.1126/science.1203543 (2011).
- 54 Hanahan, D. & Weinberg, R. A. Hallmarks of cancer: the next generation. *Cell* **144**, 646-674, doi:10.1016/j.cell.2011.02.013 (2011).
- 55 Malumbres, M. & Barbacid, M. Cell cycle, CDKs and cancer: a changing paradigm. *Nat Rev Cancer* **9**, 153-166, doi:10.1038/nrc2602 (2009).
- 56 Hortobagyi, G. N. *et al.* Ribociclib as First-Line Therapy for HR-Positive, Advanced Breast Cancer. *N Engl J Med* **375**, 1738-1748, doi:10.1056/NEJMoa1609709 (2016).
- 57 Goetz, M. P. *et al.* MONARCH 3: Abemaciclib As Initial Therapy for Advanced Breast Cancer. *J Clin Oncol* **35**, 3638-3646, doi:10.1200/JCO.2017.75.6155 (2017).
- 58 Finn, R. S. *et al.* Palbociclib and Letrozole in Advanced Breast Cancer. *N Engl J Med* **375**, 1925-1936, doi:10.1056/NEJMoa1607303 (2016).
- 59 Gaitskell, K. *et al.* Ovarian cancer survival by stage, histotype, and pre-diagnostic lifestyle factors, in the prospective UK Million Women Study. *Cancer Epidemiol* **76**, 102074, doi:10.1016/j.canep.2021.102074 (2022).
- 60 Peres, L. C. *et al.* Invasive Epithelial Ovarian Cancer Survival by Histotype and Disease Stage. *J Natl Cancer Inst* **111**, 60-68, doi:10.1093/jnci/djy071 (2019).
- 61 Anuradha, S. *et al.* Survival of Australian women with invasive epithelial ovarian cancer: a population-based study. *Med J Aust* **201**, 283-288, doi:10.5694/mja14.00132 (2014).

- 62 Leskela, S. *et al.* The Frequency and Prognostic Significance of the Histologic Type in Early-stage Ovarian Carcinoma: A Reclassification Study by the Spanish Group for Ovarian Cancer Research (GEICO). *Am J Surg Pathol* **44**, 149-161, doi:10.1097/PAS.0000000000001365 (2020).
- 63 Chiang, Y. C. *et al.* Trends in incidence and survival outcome of epithelial ovarian cancer: 30-year national population-based registry in Taiwan. *J Gynecol Oncol* **24**, 342-351, doi:10.3802/jgo.2013.24.4.342 (2013).
- 64 Li, W. *et al.* Role of exosomal proteins in cancer diagnosis. *Mol Cancer* **16**, 145, doi:10.1186/s12943-017-0706-8 (2017).
- 65 Robin, X. *et al.* pROC: an open-source package for R and S+ to analyze and compare ROC curves. *BMC Bioinformatics* **12**, 77, doi:10.1186/1471-2105-12-77 (2011).
- 66 Simpson, R. J., Kalra, H. & Mathivanan, S. ExoCarta as a resource for exosomal research. *J Extracell Vesicles* **1**, doi:10.3402/jev.v1i0.18374 (2012).
- 67 Kalra, H. *et al.* Vesiclepedia: a compendium for extracellular vesicles with continuous community annotation. *PLoS Biol* **10**, e1001450, doi:10.1371/journal.pbio.1001450 (2012).

Reviewers' Comments:

Reviewer #1:

Remarks to the Author:

The author has answered majority of my comments. However, for tumor purity, the author should provide detailed methods to describe how they evaluated the tumor purity, by histological features or estimated by proteomic data?

Reviewer #3:

Remarks to the Author:

After thoroughly examining the responses provided to the questions I initially raised during my review, I am pleased to note that the authors have addressed them adequately. Additionally, I appreciate the further edits and corrections that have been made in the manuscript. These revisions have enhanced the overall quality and clarity of the paper, and I am satisfied with the changes implemented.

Reviewer #4:

Remarks to the Author:

The authors have worked to address previous comments from reviewers and have made some improvements in the manuscript. However, there are still critical issues, especially on the analytical side, that need to be addressed:

- (1) Important clinical variables, such as Age and Stage, have been listed in Supplementary Table 1. Figure 1D shows that the stage distributions significantly differ across the three EOC subtypes. To achieve a more robust analysis, it is essential to adjust for Age and Stage in all differential and survival analyses. This adjustment should be carried out using a regression modeling framework, such as limma analysis and Cox regression, rather than relying solely on the Wilcoxon test and K-M test.
- (2) In Section 2.4, while listing the differentially expressed proteins between EOC and CT is acceptable, a more nuanced analysis is required. Specifically, differentially testing each EOC subtype versus CT would be more meaningful, given the strong heterogeneity among the three EOC subtypes. The results of these analyses can then be combined to identify both common and subtype-specific differentially expressed proteins.
- (3) For the application of WGCNA, reproducibility stands as a major concern. Combining the three EOC subtypes for analysis may not be appropriate, and this could be the reason for the large number of modules resulting from the analysis. A more suitable approach would be to perform WGCNA separately for each EOC subtype. By doing so, the analysis might be more aligned with the underlying biological variability among the subtypes and provide insights that are more robust and interpretable.
- (4) There is ambiguity in the selection process for candidate genes in Figure 6. The manuscript lacks clear information on whether CSPG4, TEM87A, and MPP7 were the top-ranked proteins in either the differential list or the survival association list. If they were, the provided p-values appear to be unadjusted, raising questions about their validity. If not, clarification is needed on the criteria used to select these three genes. Regarding MPP7, further information on its prognostic value in the two CPTAC cohorts is required. Even if MPP7 has some prognostic value, the evidence is not sufficient to conclude that it's an excellent therapeutic target based solely on the differentially expressed pattern and K-M curves.
- (5) In the last paragraph of section 3.2, the clinical relevance of discovering protein markers to differentiate between EOC and CT is unclear and needs justification.
- (6) The pathway analysis could be enriched by applying GSEA analysis for all pathways. The current analysis identifies too many significant GO terms and KEGG pathways even with an adjusted p-value cutoff of 0.01, which seems arbitrary. A more robust methodological approach would provide more meaningful insights.
- (7) The data on ProteomeXchange Consortium or OMIX is either unavailable to reviewers or missing.

This is a significant oversight that must be addressed to ensure the transparency and reproducibility of the results.

Reviewer #1 (Remarks to the Author):

Q1. The author has answered majority of my comments. However, for tumor purity, the author should provide detailed methods to describe how they evaluated the tumor purity, by histological features or estimated by proteomic data?

REPLY: We thank the reviewer for this careful consideration. We assessed tumor purity by histological features. HE staining was performed on the sections. Pathologists determine the tumor cells by histological features, such as atypia, high karyoplasmic ratio and rough chromatin. We choose the region only containing tumor cells for cutting. In addition, the cut area is appropriately reduced to ensure the exclusion of non-tumor tissue. All samples are examined by two experienced pathologists to ensure that the eligible tumor samples contain at least 90% tumor cells.

On Page4:

All of these samples were examined by two experienced pathologists who confirmed the diagnosis of the disease samples. **Eligible tumor samples contain at least 90% tumor cells.**

Reviewer #3 (Remarks to the Author):

Q1. After thoroughly examining the responses provided to the questions I initially raised during my review, I am pleased to note that the authors have addressed them adequately. Additionally, I appreciate the further edits and corrections that have been made in the manuscript. These revisions have enhanced the overall quality and clarity of the paper, and I am satisfied with the changes implemented.

REPLY: This positive feedback is very much appreciated. On behalf of the co-authors, we would like to express our great appreciation to you.

Reviewer #4 (Remarks to the Author):

The authors have worked to address previous comments from reviewers and have made some improvements in the manuscript. However, there are still critical issues, especially on the analytical side, that need to be addressed:

Q1. Important clinical variables, such as Age and Stage, have been listed in Supplementary Table 1. Figure 1D shows that the stage distributions significantly differ across the three EOC subtypes. To achieve a more robust analysis, it is essential to

adjust for Age and Stage in all differential and survival analyses. This adjustment should be carried out using a regression modeling framework, such as limma analysis and Cox regression, rather than relying solely on the Wilcoxon test and K-M test.

REPLY: We thank the reviewer for this careful consideration. Based on your suggestion, we used the limma analysis to adjust the clinical information and calculate the protein abundance differences. These results have been added to Supplementary Table 4. The findings from these analyses confirmed that most proteins (96%) that remained consistent with the original results were dysregulated patterns. In addition, we used the Cox regression analysis to adjust for age and stage and to assess the prognostic value of the proteins of interest. In the focused pathways of cell cycle, HIF-1 signaling pathway, PI3K-Akt signaling pathway, and ECM-receptor interaction, we found that the prognostic value of their proteins for overall survival and recurrence survival was more generally consistent with the original results. Results of relevant survival have been added to Supplementary Table 7. In particular, the stably expressed proteins CDKN1B (OS: P-values_(adjusting by age and stage) = 0.033, RFS: P-values_(adjusting by age and stage) = 0.010) and COL1A2 (RFS: P-values_(adjusting by age and stage) = 0.024) also had good prognostic value in age- and stage-adjusted outcomes. These consistent results highlighted the importance of the proteins and pathways which we were focusing on. Details of the revisions in the manuscript are as follows:

On Page6:

We also used 'limma' analysis to adjust for clinicopathological characteristics and calculate protein abundance differences.

On Page7:

Clinical associations of protein expression were examined using the Cox proportional hazards model, and P-values were adjusted using the False Discovery Rate (FDR) procedure. Univariable and multivariable Cox regression analysis was used to estimate the hazard ratios (HR), 95% confidence intervals (CI), Cox P-values and Cox adj.P-values of each protein.

Q2. In Section 2.4, while listing the differentially expressed proteins between EOC and CT is acceptable, a more nuanced analysis is required. Specifically, differentially testing each EOC subtype versus CT would be more meaningful, given the strong heterogeneity among the three EOC subtypes. The results of these analyses can then be

combined to identify both common and subtype-specific differentially expressed proteins.

REPLY: We thank the reviewer's suggestion. We compared the differentially expressed proteins between histological subtype and CT separately. The relevant differential proteins have been added to Supplementary Table 8. We superimposed the relative protein abundance of each subtype into the protein interaction network and identified multiple modules (such as Module31, Module7, Module18, etc.) with different regulatory roles in the three histological subtypes (Figure 4C). The protein abnormalities and functional differences in the specific modules were considered important in histological subtypes. In addition, we compared the differentially expressed proteins from EOC and histological subtypes, respectively (Figure R1). We identified multiple subtype-specific differentially expressed proteins. For example, multiple proteins in Module31 (IFIT1, STAT2, OAS2 and MX2) were all specifically highly expressed in the SC subtype. Meanwhile, we also determined common differentially expressed proteins (147 up-regulated and 518 down-regulated proteins, Figure R1), including cell cycle proteins (CDK4, CDK6, CDKN1B) and ECM proteins (COL1A2, COL6A1, COL6A2, COL6A3), suggesting a general disruption of these biological functions in EOC.

Figure R1. Venn diagrams of the differentially expressed proteins in EOC, CCC, EC, and SC.

On Page6:

In addition, we compared the differentially expressed proteins between histological subtype and CT separately using the above methods.

On Page13:

Subsequently, we evaluated the abnormal expression levels of the proteins by comparing each histological subtype with CT separately (Supplementary Figure 3D and

Supplementary Table 8). When the relative protein abundances per pathogenic category were superimposed overlaid on the network, multiple modules (such as Module31, Module7, Module18, etc.) with differential regulation among the three histological subtypes were identified (Figure 4C). The protein abnormalities and functional variations in the specific modules were considered essential.

Q3. For the application of WGCNA, reproducibility stands as a major concern. Combining the three EOC subtypes for analysis may not be appropriate, and this could be the reason for the large number of modules resulting from the analysis. A more suitable approach would be to perform WGCNA separately for each EOC subtype. By doing so, the analysis might be more aligned with the underlying biological variability among the subtypes and provide insights that are more robust and interpretable.

REPLY: We thank the reviewer for this careful consideration. Based on your suggestion, we used WGCNA to analyze each histological subtype separately. The results showed that 40, 65, and 37 modules were recognized in the CCC, EC, and SC subtypes, respectively. The number of histological modules was not significantly reduced compared to those obtained from EOC modules (58 modules). In addition, we attempted to gain a more comprehensive understanding of the overlap between the EOC modules and the histological subtype modules at the network level, in which the protein association of each module was assessed using the hypergeometric Fisher's exact test (adj.P-values < 0.05). The results showed that almost all histological subtype modules (138/142, 97%) had a statistically significant overlap with the EOC modules (Supplementary Figure 3C). Of these, the four modules without significant overlap were mainly of the EC subtype, including Module13, Module38, Module55, and Module64. The main reason for these non-overlapping modules may be that fewer proteins (≤ 10 proteins) were co-expressed within the modules. These results may indicate that the EOC modules have a better overlap with the histological subtype modules, and have a certain degree of reproducibility. Details of the revisions in the manuscript are as follows:

On Page13:

In addition, we examined the overlap between the above EOC modules and the histological subtype modules, where the protein associations between the modules were assessed using the hypergeometric Fisher's exact test (adj. P-values < 0.05). The results

showed that almost all histological subtype modules (97%) had statistically significant overlap with the EOC modules (Supplementary Figure 3C), demonstrating a better overlap between the EOC modules and the histological subtype modules.

Supplementary Figure 3C. Degree of overlap between proteins in the EOC modules and the histological subtype modules.

Q4. *There is ambiguity in the selection process for candidate genes in Figure 6. The manuscript lacks clear information on whether CSPG4, TMEM87A, and MPP7 were the top-ranked proteins in either the differential list or the survival association list. If they were, the provided p-values appear to be unadjusted, raising questions about their validity. If not, clarification is needed on the criteria used to select these three genes. Regarding MPP7, further information on its prognostic value in the two CPTAC cohorts is required. Even if MPP7 has some prognostic value, the evidence is not sufficient to conclude that it's an excellent therapeutic target based solely on the differentially expressed pattern and K-M curves.*

REPLY: We thank the reviewer's suggestion. The screening criteria for candidate proteins of Result3.6 were not clearly described in the previous manuscript. The detailed screening criteria are listed below: ①Differentially expressed proteins identified by comparison of EOC samples and CT samples (from Result3.2); ②Subtype-specific proteins identified by comparison among the three histological subtypes (from Result3.5); ③Proteins with prognostic value in histological subtype screened based on Kaplan-Meier curves and Cox regression analysis; ④Proteins associated with overall survival in only one histological subtype, not in all two or three

histological subtypes. The candidate proteins for each histological subtype were required to meet all the above conditions. Therefore, we have selected CSPG4, TMEM87A, and MPP7 as candidate markers. Relevant details have been revised in the manuscript.

Based on your suggestion, we analyzed the prognostic value of MPP7 in the CPTAC cohort and confirmed that high expression of MPP7 was associated with poor overall survival in serous carcinoma patients (log-rank P-values = 0.02). The expression pattern and prognostic value of MPP7 are the basis for measuring it as a potential therapeutic target. We also performed in vitro experiments to validate the biological function of MPP7 in serous carcinoma cells. CCK-8 assays showed that shRNA-mediated MPP7 knockdown decreased the proliferation of serous carcinoma cells (Supplementary Figure 7A). Flow cytometric analysis of cell cycle demonstrated that MPP7 knockdown resulted in a decrease of G1-S transition in serous carcinoma cells (Figure 6G). Flow cytometric analysis of cell apoptosis revealed that apoptosis of serous carcinoma cells was induced by MPP7 knockdown (Figure 6H). Transwell assays showed that migration and invasion of serous carcinoma cells were inhibited by MPP7 knockdown (Supplementary Figure 7B-C). These results highlighted the important role of MPP7 in SC. Protein MPP7 as a potential therapeutic target might provide new perspectives for targeted SC therapy, and more comprehensive investigation and validation in the future would provide a better overall view and promote new research directions for SC patients. Details of the revisions in the manuscript are as follows:

On Page15:

We tried to find potential drug targets for each histological subtypes. Firstly, the intersection of differentially expressed proteins in EOC and CT with subtype-specific proteins was used as a candidate protein list. Then, independent and significant prognostic proteins were identified for each histological subtype based on Kaplan-Meier curves and Cox regression analysis.

On Page16:

The prognostic value of MPP7 was also validated in the CPTAC dataset (log-rank P-values < 0.05).

Q5. In the last paragraph of section 3.2, the clinical relevance of discovering protein markers to differentiate between EOC and CT is unclear and needs justification.

REPLY: We thank the reviewer for this careful consideration. The clinical relevance of the protein markers of Result3.2 was not clearly described in the previous manuscript. We aimed to measure the ability of differentially expressed proteins to classify EOC samples and CT samples. Therefore, the classification performance of each protein was evaluated using ROC curves and validated in the CPTAC cohort to obtain multiple upregulated proteins with the potential ability to discriminate between EOC and CT samples. Furthermore, we integrated exosome protein lists from the ExoCarta and Vesiclepedia databases, and focused on their overlap with the above proteins. We found that these partial proteins with the potential ability to discriminate between EOC and CT samples could be detected in exosomes. Since cancer exosomal proteomic research is currently in its infancy, we do not currently have access to exosome protein expression profiles for ovarian cancer to determine the function and value of the above proteins as exosomes. In future work, we would like to perform a more detailed analysis of the exosomal proteins identified in this study to explore the possibility of circulating proteins as diagnostic markers for EOC, providing new perspectives for EOC diagnosis. Details of the revisions in the manuscript are as follows:

On Page11:

We aimed to measure the ability of differentially expressed proteins to classify EOC samples and CT samples. Firstly, to distinguish between tumor samples and normal samples, differentially expressed proteins were used as the initial feature. Then, the classification performance of each feature was evaluated using the area under curve (AUC) of the receiver operating characteristic (ROC) curves (R package 'pROC'). Additionally, proteins with good discriminatory power were enrolled for further validation in the ovarian cancer cohort of McDermott, J. E. et al. (Clinical Proteomic Tumor Analysis Consortium, CPTAC cohort). We observed that multiple up-regulated proteins exhibited strong discriminatory ability with a mean AUC greater than 0.9 (current cohort) and further validated in the CPTAC cohort¹⁰ with a mean AUC also greater than 0.8. The exosome protein lists from the ExoCarta (<http://www.exocarta.org/>) and Vesiclepedia (<http://microvesicles.org/>) databases. Proteins with discriminatory ability have overlapping portions with exosomal proteins (Figure 2E), and these overlapping proteins may provide new perspectives for EOC research.

Q6. The pathway analysis could be enriched by applying GSEA analysis for all pathways. The current analysis identifies too many significant GO terms and KEGG pathways even with an adjusted p-value cutoff of 0.01, which seems arbitrary. A more robust methodological approach would provide more meaningful insights.

REPLY: We thank the reviewer's suggestion. Based on your comments, we used GSEA to re-analyze the GO terms and KEGG pathways that might be affected by the differentially expressed proteins (Supplementary Table 6). The KEGG pathways enriched by GSEA analysis were generally consistent with the previous enrichment results. In particular, both the enriched TOP pathways (such as peroxisome, lysosome, focal adhesion, drug metabolism, and glutathione metabolism, Figure 2C) and the pathways of special interest (such as DNA replication and ECM-receptor interaction, Figure 3) were consistent with the GSEA analysis results. Similarly, the significantly enriched GO terms showed generally consistent results in both analyses. For example, the results of GSEA analysis were also mainly involved in immune response, DNA repair/damage, development and cell death, including humoral immune response, negative regulation of proteolysis, regulation of endocytosis, and so on (Figure 2B). The consistent results of the two analyses provide some insight into the biological functions in which the differentially expressed proteins may be involved. Details of the revisions in the manuscript are as follows:

On Page10:

Meanwhile, the GSEA-enriched GO biological processes and KEGG pathways also highlighted consistent biological functions (Supplementary Table 6).

Q7. The data on ProteomeXchange Consortium or OMIX is either unavailable to reviewers or missing. This is a significant oversight that must be addressed to ensure the transparency and reproducibility of the results.

REPLY: We thank the reviewer for this careful consideration. Our data have been uploaded to the public repository. The mass spectrometry proteomic data were deposited to the ProteomeXchange Consortium (<http://www.proteomexchange.org/>) via the PRIDE partner repository with the identifier PXD033741. In addition, we obtained the necessary approval from the Chinese Ministry of Science and Technology (MOST, <https://grants.most.gov.cn/grants/>) related to export the genetic information and materials associated with this work (registration number: 2023BAT0). The mass spectrometry proteomic data also have been deposited in the Open Archive for

Miscellaneous Data (OMIX, <https://ngdc.cncb.ac.cn/omix/>) with the identifier OMIX002719.

This data will be made public after the article has been published. Please visit the PRIDE Archive login page <https://www.ebi.ac.uk/pride/login> to enter the reviewer credentials and access the data.

Username: reviewer_pxd033741@ebi.ac.uk Password: j5tuiw9s

On Page20:

Data and materials availability

The mass spectrometry proteomic data have been deposited to the ProteomeXchange Consortium (<http://www.proteomexchange.org/>) via the PRIDE partner repository with the identifier PXD033741. The mass spectrometry proteomic data also have been deposited to the Open Archive for Miscellaneous Data (OMIX, <https://ngdc.cncb.ac.cn/omix/>) with the identifier OMIX002719.

We have tried our best to improve the manuscript and have made some changes, marked in red in the revised paper which will not affect the content and framework of the paper. We sincerely appreciate the hard work of the reviewers and hope that the correction will meet with approval. Thank you again for your comments and suggestions.

Reviewers' Comments:

Reviewer #4:

Remarks to the Author:

The authors have satisfactorily addressed my concerns, and I believe the manuscript is now ready for publication

REVIEWER COMMENTS

Reviewer #4 (Remarks to the Author):

The authors have satisfactorily addressed my concerns, and I believe the manuscript is now ready for publication.

REPLY: Thank you very much for your positive feedback. On behalf of the co-authors, we would like to express our great appreciation to you.